# Targeted anatomical and functional identification of antinociceptive and pronociceptive serotonergic neurons that project to the spinal dorsal horn

Robert Philip Ganley[1]*, Marilia Magalhaes de Sousa[1], Kira Werder[1], Tugce Öztürk[1], Raquel Mendes[1], Matteo Ranucci[1], Hendrik Wildner[1], Hanns Ulrich Zeilhofer[1,2]

[1]Institute for Pharmacology and Toxicology, University of Zurich, Zurich, Switzerland; [2]Institute of Pharmaceutical Sciences, Swiss Federal Institute of Technology, Zurich, Switzerland

**Abstract** Spinally projecting serotonergic neurons play a key role in controlling pain sensitivity and can either increase or decrease nociception depending on physiological context. It is currently unknown how serotonergic neurons mediate these opposing effects. Utilizing virus-based strategies and Tph2-Cre transgenic mice, we identified two anatomically separated populations of serotonergic hindbrain neurons located in the lateral paragigantocellularis (LPGi) and the medial hindbrain, which respectively innervate the superficial and deep spinal dorsal horn and have contrasting effects on sensory perception. Our tracing experiments revealed that serotonergic neurons of the LPGi were much more susceptible to transduction with spinally injected AAV2retro vectors than medial hindbrain serotonergic neurons. Taking advantage of this difference, we employed intersectional chemogenetic approaches to demonstrate that activation of the LPGi serotonergic projections decreases thermal sensitivity, whereas activation of medial serotonergic neurons increases sensitivity to mechanical von Frey stimulation. Together these results suggest that there are functionally distinct classes of serotonergic hindbrain neurons that differ in their anatomical location in the hindbrain, their postsynaptic targets in the spinal cord, and their impact on nociceptive sensitivity. The LPGi neurons that give rise to rather global and bilateral projections throughout the rostrocaudal extent of the spinal cord appear to be ideally poised to contribute to widespread systemic pain control.

**\*For correspondence:**
robert.ganley@pharma.uzh.ch

**Competing interest:** The authors declare that no competing interests exist.

## Editor's evaluation

The study reveals anatomically and functionally distinct classes of serotonergic hindbrain neurons that are distinguished by their postsynaptic targets in the spinal cord as well as their contributions to behavioral responses to painful stimuli. These findings advance our understanding of the descending modulation of sensory processing in the spinal cord dorsal horn.

## Introduction

Descending pain control is a critical endogenous mechanism of pain modulation that is required for survival (*Millan, 2002*), and allows an organism to respond in an appropriate context-dependent manner to external threats. Key brain structures involved in this system were discovered through stimulation-produced analgesia experiments and include the periaqueductal grey matter (PAG) and the rostroventromedial medulla (RVM) of the hindbrain (*Basbaum and Fields, 1984*). This descending

system acts to inhibit the flow of nociceptive information through the spinal dorsal horn and requires descending tracts within the dorsolateral funiculus to mediate its effects (*Basbaum et al., 1976*; *Basbaum et al., 1977*). The RVM is also required for maintaining some chronic pain states through a process of descending facilitation (*Porreca et al., 2001*; *Zhang et al., 2009*), indicating that this area has a bidirectional control over pain sensitivity. Both descending inhibition and facilitation will ultimately require the activation of descending tracts to modulate the neuronal activity within the dorsal horn. To understand the precise neuronal circuitry and transmitter systems underlying descending pain inhibition and facilitation, an understanding of the different projection neurons from the RVM to the spinal cord is needed.

Monoamines are important neurotransmitters involved in descending pain control, with serotonin (5-HT) being able to inhibit nociception when injected intrathecally (*Wang, 1977*). Spinal 5-HT is also known to contribute to certain forms of endogenous pain suppression, such as stress-induced analgesia (*Yesilyurt et al., 2015*) but is also required to maintain chronic pain in rodent models of nerve injury (*Wei et al., 2010*). This pain facilitation in pre-clinical neuropathy models is thought to be a result of reduced tonic diffuse noxious inhibitory controls, due to increased spinal 5-HT$_3$ receptor activation (*Bannister and Dickenson, 2016*; *Bannister et al., 2015*). The variety and abundance of 5-HT receptors within the spinal dorsal horn likely explains, at least in part, why these multiple effects are observed (*Bardoni, 2019*). Clearly, spinal 5-HT and the serotonergic projections to this region have a complex and bidirectional control over pain perception that require further study.

Functional studies of specific descending projections have been facilitated by the development of viral retrograde tracers, such as AAV2retro and canine adenoviruses (*François et al., 2017*; *Hirschberg et al., 2017*; *Tervo et al., 2016*). However, it has been reported that the AAV2retro serotype, may not infect serotonergic hindbrain neurons as efficiently as other descending projections (*Tervo et al., 2016*; *Wang et al., 2018*). Further, despite the dense serotonergic innervation of the spinal dorsal horn, studies employing transsynaptic neuronal circuit tracing from dorsal horn neurons with modified rabies viruses rarely found labeled hindbrain serotonergic neurons (*François et al., 2017*; *Liu et al., 2019*). This is surprising since both descending serotonergic neurons and dorsal horn interneurons are known to strongly influence nociception, and it is possible that these rabies-virus-based tracers fail to detect functional connections between serotonergic neurons and starter populations within the spinal cord. Together these reports would suggest that descending serotonergic projections of the hindbrain are particularly challenging to study using the currently available tools.

In an attempt to address this discrepancy, we compared the labeling efficiency of viral and non-viral tracers. Specifically, we examined the susceptibility of serotonergic hindbrain neurons to retrograde spinal transduction by AAV2retro serotype vectors (*Tervo et al., 2016*). Additionally, we traced serotonergic neurons of the hindbrain with modified rabies viruses and tested whether they could be labeled from dorsal horn neuron starter populations with transsynaptic rabies tracing. To allow functional interrogation of descending serotonergic neurons, we assessed the specificity of the Tph2-Cre mouse line and used this together with the preferential transduction efficiency of AAV2retro to develop an intersectional system that allowed selective labeling and manipulation of serotonergic neurons in the lateral paragigantocellularis (LPGi), or the medial serotonergic neurons including the nucleus raphe magnus (NRM). We find that the lateral and medial serotonergic neurons of the RVM are distinct in terms of their anatomical organization, susceptibility to AAV2retro transduction, and influence on acute nociception.

## Results

### Spinal injection of AAV2retro serotype vectors preferentially transduce TPH2-containing projection neurons in the lateral hindbrain

AAV vectors have been developed to permit the efficient retrograde labeling of projection neurons via their axon terminals (*Tervo et al., 2016*). However, it is debatable whether these vectors are capable of transducing serotonergic neurons that project to the spinal cord (*Wang et al., 2018*). To test this, we used AAV2retro serotype vectors to retrogradely label hindbrain neurons that project to the spinal cord to assess their ability to transduce serotonergic projection neurons (*Figure 1A*). Following intraspinal injection of AAV2retro.GFP, many transduced neurons were labeled within the RVM (*Figure 1B*). When we inspected the location of neurons that expressed both TPH2 and eGFP

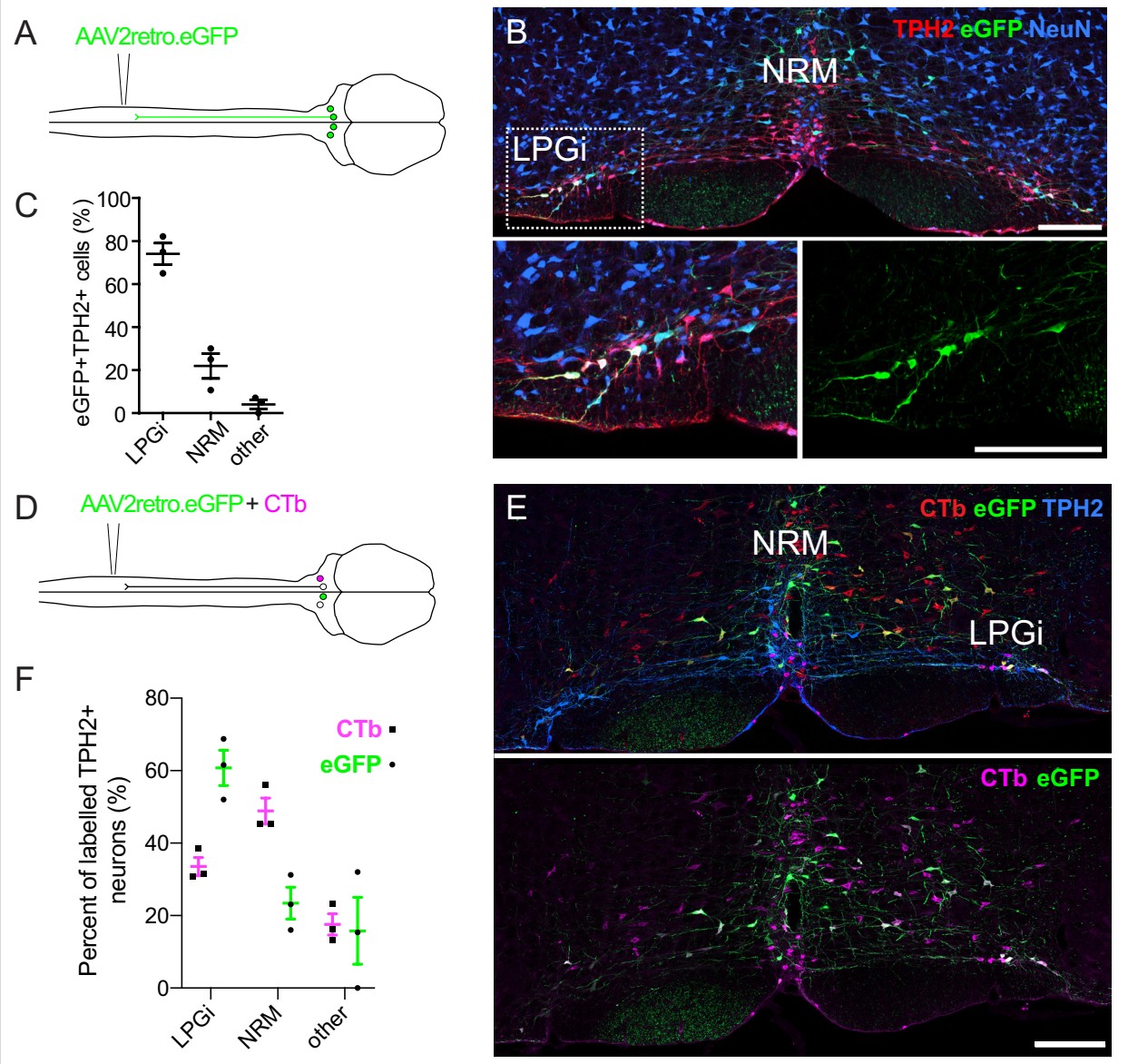

**Figure 1.** Retrograde labeling of spinally projecting serotonergic neurons with AAV2retro vectors and Cholera toxin b subunit. (**A**). Injection scheme for retrograde labeling of spinally projecting neurons with AAV2retro.eGFP. (**B**) Image of the ventral hindbrain containing eGFP-labeled neurons (scale bar = 200 μm). Inset shows enlargement of the LPGi to reveal eGFP neurons that also express TPH2 (scale bar = 200 μm). (**C**). Quantification of cell location for eGFP-labeled neurons that express TPH2, each datapoint is a count per animal (n=3) (**D**). Injection scheme for retrograde tracing from the spinal dorsal horn with AAV2retro.eGFP and CTb. (**E**). Representative image of the ventral hindbrain containing CTb-labeled and AAV2retro-transduced projection neurons (scale bar = 200 μm). (**F**). Anatomical locations of retrogradely labeled TPH2 +hindbrain neurons labeled with CTb or AAV2retro (n=3 animals).

The online version of this article includes the following figure supplement(s) for figure 1:

**Figure supplement 1.** Defined areas in the RVM used for quantifying the location of retrogradely labeled neurons.

**Figure supplement 2.** Retrograde labeling of serotonergic hindbrain neurons with CTb.

**Figure supplement 3.** Proportion of neurons retrogradely labeled with CTb and AAV2retro in different RVM areas.

**Figure supplement 4.** labeling of spinally projecting serotonergic neurons with direct rabies infection and transsynaptic rabies tracing.

(according to scheme depicted in *Figure 1—figure supplement 1*), we found that the majority of these cells were located within the lateral paragigantocellularis (LPGi) (*Figure 1B*). Quantification of the location of the eGFP-labeled TPH2-expressing neurons indicated that 74% were in the LPGi both ipsilateral and contralateral to the spinal cord injection site (left hand side). In agreement with (*Wang et al., 2018*), far fewer eGFP + TPH2+neurons were found within the midline serotonergic

**Table 1.** Quantification of hindbrain neurons traced from the spinal dorsal horn with AAV2retro.eGFP and CTb.
The number in each column is the total number of cells counted for each group, with the range of cells counted for each animal in parentheses. For percentages, the average value is presented with the range for each animal given in parentheses.

| Brain region | Subset | GFP only | Ctb only | GFP +Ctb + | %GFP only | %Ctb only | %GFP +Ctb + |
|---|---|---|---|---|---|---|---|
| RVM (all neurons) | - | 99 (11–57) | 679 (149–319) | 279 (65–134) | 9.1 (4.0–12.1) | 65.6 (59.7–73.3) | 25.3 (22.7–28.1) |
| | All | 5 (0–4) | 269 (63–123) | 62 (12–25) | 1.5 (0–5.1) | 80.1 (76.1–83.1) | 18.5 (15.2–22.9) |
| | LPGi | 5 (0–4) | 79 (16–44) | 35 (7–15) | 4.2 (0–14.8) | 66.4 (54.3–77.2) | 29.4 (22.8–42.9) |
| | NRM | 0 (0–0) | 143 (37–63) | 15 (4–6) | 0 (0–0) | 90.5 (87.8–94.0) | 9.5 (6.0–12.2) |
| RVM (TPH2 +neurons) | Other | 0 (0–0) | 47 (10–21) | 12 (0–8) | 0 (0–0) | 79.7 (66.7–100) | 20.3 (0–33.3) |

nuclei 23/100 neurons in NRM, (n=3 animals; *Figure 1C*). However, most non-serotonergic projection neurons (eGFP + TPH2-) were found close to the midline of the ventral hindbrain 412/439 neurons in midline, (n=3 animals; see *Figure 1B* and *Figure 1E*).

There is increasing evidence to suggest that some projections are resistant to retrograde transduction with AAV2retro vectors, and comparisons with independent tracing methods are required to demonstrate this resistance (*Ganley et al., 2021*; *Tervo et al., 2016*). To study the anatomical organization of serotonergic pathways using a non-viral method, we used Cholera toxin b subunit (CTb) retrograde tracing and compared the hindbrain labeling with our AAV2retro tracing experiments (*Figure 1—figure supplement 2A*).

Strikingly, many CTb-labeled TPH2-expressing hindbrain neurons were found in the midline NRM, although CTb-containing TPH2-expressing neurons were also found in the LPGi (*Figure 1—figure supplement 2B*). The location of these CTb + TPH2 + cells was more evenly divided between the NRM and LPGi that the eGFP + TPH2 + cells from the AAV2retro labeling (*Figure 1—figure supplement 2C* compared to *Figure 1C*). To directly test whether there were differences in labeling efficiency between AAV2retro and CTb, AAV2retro.GFP and CTb were co-injected into the spinal dorsal horn (*Figure 1D*). In total, more neurons in the ventral hindbrain were retrogradely labeled with CTb than AAV2retro.eGFP (*Figure 1—figure supplement 3A* and *Table 1*). When only TPH2-expressing retrogradely labeled neurons were examined, far fewer were labelled with AAV2retro.GFP relative to the overall population of traced neurons (*Figure 1—figure supplement 3* and *Table 1*). However, when the retrogradely labeled TPH2 + neurons were divided into separate areas, it was apparent that a lower percentage of neurons in the medial serotonergic nuclei including the NRM were labeled with eGFP compared to the LPGi, whereas the proportion of serotonergic neurons in the LPGi labeled with AAV2retro, CTb, and both AAV2retro and CTb were similar to the general population of retrogradely labeled hindbrain neurons (*Figure 1—figure supplement 3A, B*). Therefore, we conclude that the reduced labeling of hindbrain serotonergic neurons with AAV2retro was due to a resistance of midline serotonergic neurons to AAV2retro-mediated transduction. In contrast, laterally located serotonergic neurons were more amenable to retrograde transduction with AAV2retro. A summary of these data can be found in *Table 1*.

### Serotonergic hindbrain neurons are rarely infected using transsynaptic rabies tracing from the dorsal horn

Since many serotonergic neurons were largely resistant to AAV2retro transduction, we considered that these may be resistant to other viruses used for retrograde and neuronal circuit tracing. Transsynaptic rabies tracing is a commonly used technique for tracing neuronal circuits from genetically defined populations of neurons and provides useful information regarding the presynaptic inputs to a neuronal population. (*Wickersham et al., 2007b*). However, in the case of serotonergic neurons, the results of previous studies have demonstrated that very few neurons of the hindbrain are traced from dorsal horn starter populations, despite the dense serotonergic innervation of the superficial dorsal horn and the important role of serotonin in pain modulation (*François et al., 2017*; *Liu et al., 2019*). This suggests that serotonergic neurons are either traced with only limited efficacy, or that there is highly specific serotonergic innervation of some spinal neuron populations but not others.

**Table 2.** Neurons in the ventral hindbrain traced from the spinal cord with modified rabies viruses.
Number represents the total number of cells counted with the range of cells counted for each animal in parentheses. Percentages of labeled neurons expressing TPH2 are the average for all cells counted per group, with the range for each animal given in parentheses.

| Labeling strategy | Rabies virus | Animals | Total GFP + cells counted in RVM | GFP neurons expressing TPH2 | GFP neurons not expressing TPH2 | % GFP +neurons that are TPH2+ |
|---|---|---|---|---|---|---|
| Direct infection (5d) | SAD.RabiesΔG-eGFP (SAD-G) | 3 | 25 (5–11) | 11 (2–5) | 14 (3–6) | 44 (40–45) |
| Direct infection (7d) | SAD.RabiesΔG-eGFP (SAD-G) | 3 | 36 (7–17) | 4 (1-2) | 32 (6–16) | 11 (6–16) |
| Monosynaptic tracing | SAD.RabiesΔG-eGFP (EnvA) | 4 | 33 (3–20) | 1 (0–1) | 32 (3–19) | 3 (0–5) |

To test whether serotonergic hindbrain neurons are susceptible to rabies virus infection, we injected a rabies virus, whose genome lacked the glycoprotein-encoding sequence but was pseudotyped with the SAD glycoprotein (SAD.RabiesΔG-eGFP (SAD-G)), into the spinal dorsal horn (*Figure 1—figure supplement 4A*). This virus can directly infect most neurons but cannot be propagated beyond the initially infected neurons due to the lack of the rabies glycoprotein required for transsynaptic spread (*Albisetti et al., 2017*; *Wickersham et al., 2007a*). Five days after injection of SAD.rabΔG-eGFP (SAD-G), 44% of the labeled hindbrain neurons contained detectable TPH2 immunoreactivity (*Figure 1—figure supplement 4B, C*). Notably, the percentage of labeled neurons dropped to only 12% of labeled hindbrain neurons at day 7 post injection (*Figure 1—figure supplement 4C*). This reduction may be a result of rabies virus toxicity, which could lead to a downregulation of cytoplasmic enzymes such as TPH2 to undetectable levels, and hence an underestimation of labeled serotonergic neurons. For this reason, we used a 5-day survival time for all subsequent experiments involving rabies viruses.

To determine whether serotonergic hindbrain neurons could be traced transsynaptically from the spinal dorsal horn, we used Hoxb8-Cre mice to define a broad starter population of spinal cord neurons. During development, Hoxb8-Cre is expressed transiently in almost all spinal neurons and astrocytes caudal to C4. It can be used to define a starter population that includes virtually all dorsal horn neurons (*Witschi et al., 2010*). We used Hoxb8-Cre; ROSA[TVA] mice to induce stable expression of TVA in most dorsal horn neurons during development, enabling their infection in the adult with EnvA pseudotyped rabies viruses (SAD.RabiesΔG-GFP (EnvA)). To allow transsynaptic spread from the starter population, a helper virus containing mCherry and rabies glycoprotein was injected two weeks prior to injection of SAD.RabiesΔG-GFP (EnvA) (*Figure 1—figure supplement 4D*).

Five days after rabies virus injection, neurons in the ventral hindbrain labeled with eGFP were assessed for TPH2 expression by immunostaining. On average, the number of eGFP-labeled hindbrain neurons counted per animal was similar between direct and transsynaptic tracing from the spinal cord (8.3 and 8.25, respectively), both of which had far lower labeling efficiency when compared to labeling with either CTb or AAV2retro labeling. We only observed coexpression of eGFP and TPH2 once in a total of 33 inspected neurons, corresponding to 3% of our total sample (*Figure 1—figure supplement 4E, F*). A summary of all rabies virus tracing experiments can be found in *Table 2*. In the hindbrain, we never observed colocalization of mCherry and eGFP, suggesting that all eGFP-expressing neurons in the hindbrain were transsynaptically traced from the dorsal horn. Together these data demonstrate that although TPH2-expressing hindbrain neurons can be directly infected with modified rabies viruses, they are largely underrepresented with transsynaptic tracing from the dorsal horn. This may partly explain their absence from many of the circuit tracing studies that have used this approach.

## Descending serotonergic projection neurons of the LPGi can be preferentially labeled using AAV2retro vectors and Tph2-Cre mouse line

In order to study descending serotonergic neurons functionally, we must first identify a suitable method to specifically influence those neurons. TPH2-containing neurons of the LPGi were susceptible to retrograde transduction by AAV2retro in contrast to other serotonergic nuclei of the hindbrain

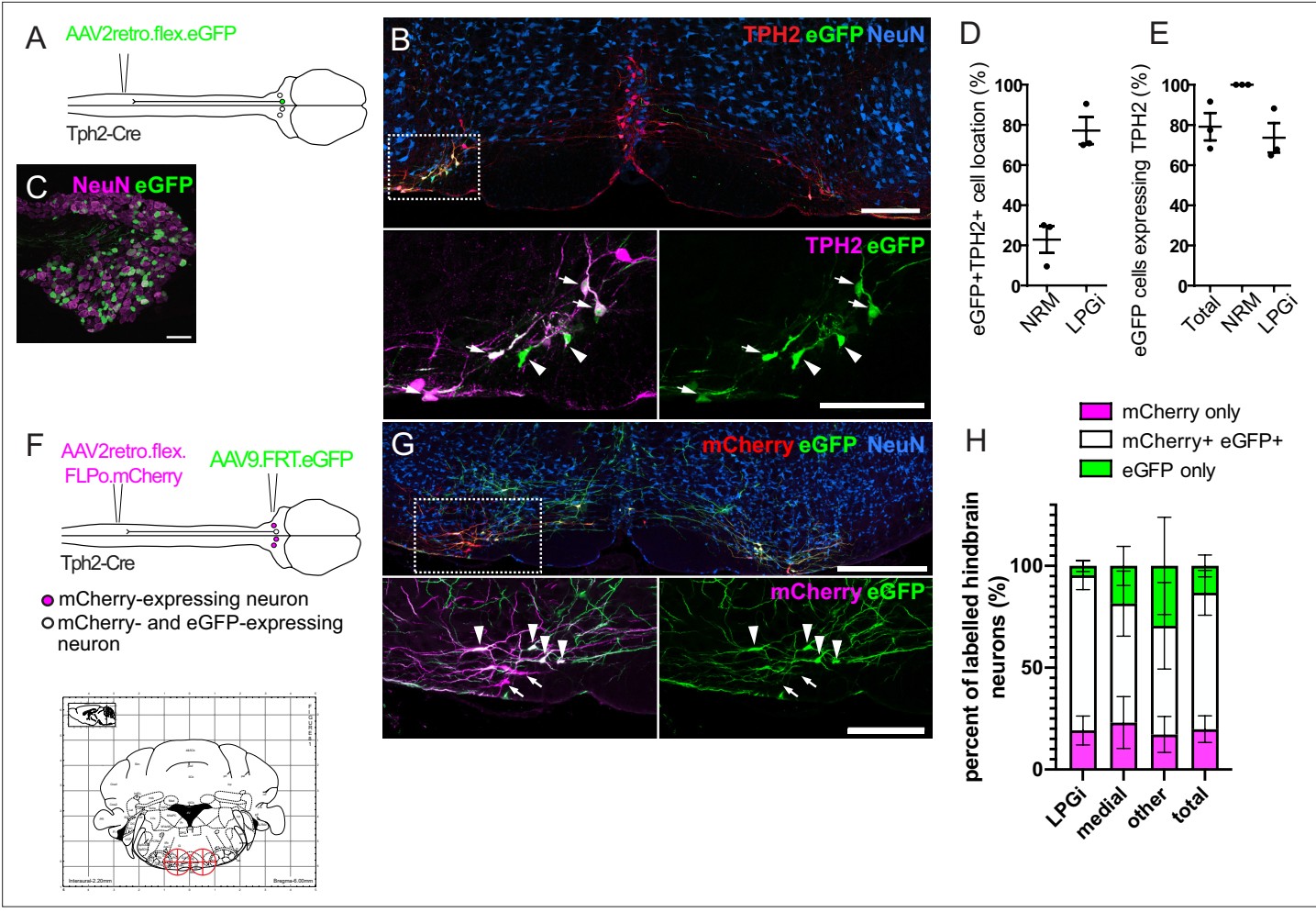

**Figure 2.** Labeling of spinally projecting neurons in the Tph2-Cre mouse with AAV2retro vectors. (**A**). Injection scheme for labeling spinally projecting Tph2-Cre neurons. (**B**). An example of a hindbrain section containing neurons labeled with eGFP (scale bar = 200 µm). Inset is an enlargement of the area indicated in the dashed box, with many eGFP-labeled cells found to express detectable TPH2 that are indicated with arrows. Cells that were labelled with eGFP not containing detectable TPH2 are indicated with arrowheads (scale bar = 200 µm). (**C**). Image of an ipsilateral DRG from a Tph2-Cre mouse that received a spinal injection of AAV2retro.flex.eGFP, showing many eGFP-expressing neurons (scale bar = 100 µm). (**D**). Quantification of the location of eGFP-labeled cells in the hindbrain. (**E**). Quantification of the hindbrain neurons labeled with eGFP that also contain TPH2. For **D**. and **E**. each datapoint is a count per animal (n=3 animals). (**F**). Injection scheme for the intersectional labeling of spinally projecting Tph2-Cre neurons. Brain injection coordinates (−6,+/-0.5, 5.9) from bregma, for the labeling of spinally projecting Tph2-Cre neurons in the LPGi, according to the mouse brain atlas. Target injection sites are indicated in red crosshairs. (**G**). Example of a hindbrain section from a Tph2-Cre mouse that received the injections illustrated in **F** (scale bar = 500 µm). Inset is an enlargement of the boxed area and highlights neurons that were captured with the brain injection and express eGFP (arrowheads) as well as neurons that were directly labeled from the spinal cord injection (mCherry+) that were not transduced from the hindbrain injection (arrows),(scale bar = 200 µm). H. Quantification of the mCherry-expressing cells that are labeled with eGFP, which indicated that they were captured with the hindbrain injection. The percentages of mCherry-only and eGFP-only cells are also quantified for each hindbrain area that contains serotonergic neurons, and areas that could not be assigned as either medial or LPGi were classified as 'other' (n=4 animals).

(*Figure 2*), which would permit selective transduction of these neurons using AAV2retro. Further, mouse lines expressing Cre recombinase within defined neuronal populations have been widely and successfully used to functionally study a variety of spinal neuron subtypes (*Foster et al., 2015*; *Gatto et al., 2021*; *Huang et al., 2018*; *Peirs et al., 2021*). We therefore assessed the reliability of the Tph2-Cre mouse line as a marker of serotonergic neurons and a potential tool to gain genetic access to these cells. To determine whether the Cre expression in this mouse line was specific to serotonergic neurons that project to the spinal cord, we injected the lumbar dorsal horn of Tph2-Cre mice with an AAV2retro serotype vector containing a Cre-dependent eGFP construct (*Figure 2A*). This resulted in TPH2-expressing neurons of the hindbrain being labeled with eGFP, particularly in the LPGi (*Figure 2B*). In addition, many neurons were labeled in the ipsilateral DRG (*Figure 2C*). We observed

**Table 3.** Cell counts for eGFP-labeled cells in the hindbrain from injection of AAV2retro.flex.eGFP into the spinal cord of Tph2-Cre animals (n=3) the total number of cells counted is indicated and the range of cells counted per animal is indicated in parentheses.

| GFP (all) | GFP (NRM) | GFP (LPGi) | GFP +TPH2+ (all) | GFP +TPH2+ (NRM) | GFP +TPH2+ (LPGi) |
|---|---|---|---|---|---|
| 127 (24–63) | 25 (7–12) | 102 (17–57) | 96 (22–43) | 25 (7–12) | 71 (15–37) |

a similar pattern in cell location as in the AAV2retro.eGFP labeling experiments (*Figure 1C*), with 72% of eGFP neurons being present bilaterally in the LPGi (*Figure 2D* and *Table 3*). Furthermore, 77.5% of all neurons labeled with AAV2retro.flex.eGFP contained detectable levels of TPH2 (*Figure 2E* and *Table 3*). Therefore, among the hindbrain neurons that project to the spinal dorsal horn Tph2-Cre is expressed specifically in serotonergic neurons, but may also be expressed in other neurons of the nervous system, including a subset of DRG neurons.

To specifically manipulate spinally projecting serotonergic neurons without influencing other neuronal populations, such as those in the DRG, we used an intersectional strategy to induce reporter expression in spinally projecting Tph2-Cre neurons of the hindbrain (*Figure 2F*). This strategy utilized AAV2retro serotype vectors containing a Cre-dependent optimized flippase (FLPo), which, when injected intraspinally, results in FLPo expression in Cre-expressing neurons that project to the dorsal horn. The AAV2retro serotype would also enable expression restricted to laterally located serotonin-expressing neurons. This construct also contained the coding sequence for mCherry, allowing these transduced cells to be visualized (AAV2retro.flex.FLPo.mCherry). The hindbrain was injected one week later with an AAV containing a flippase-dependent eGFP viral vector (AAV9.FRT.eGFP), to capture the neurons that were transduced from the spinal cord with AAV2retro (*Figure 2F and G*). This approach enables the quantification of intersectionally labeled serotonergic neurons (mCherry- and eGFP-expressing), relative to the population of neurons directly labeled from the spinal dorsal horn (mCherry-expressing). Therefore, the accuracy of the hindbrain injections and the proportion of the descending projection neurons captured within the injection site could be determined.

Different brain injection coordinates were tested to optimize the proportion of spinally projecting Tph2-Cre neurons that were captured with the stereotaxic brain injection. Bilateral injections of AAV9.FRT.eGFP (−6, ±0.5, −5.9 relative to Bregma) was the most efficient in terms of mCherry +eGFP + neurons labeled in the hindbrain (*Figure 2F, G and H*). With this approach, we labeled 80.6% of all retrogradely transduced neurons with eGFP with the remaining 19.4% containing mCherry only, indicating that most retrogradely transduced neurons are labeled using this strategy (*Figure 2H* and *Table 4*). Additionally, this proportion was similar between the different areas containing retrogradely labeled serotonergic neurons (LPGi = 80.6%, Medial = 78.3%, other = 82.4%, labeled neurons containing eGFP), suggesting these injection sites encompass most regions that contain AAV2retro-labeled serotonergic projection neurons. Some cells only expressed eGFP without detectable mCherry staining, either due to non-specific recombination of the FRT sites or low to undetectable levels of mCherry present in these neurons. We found that a smaller proportion of eGFP-only neurons were present in the LPGi compared to the medial serotonergic nuclei (4.6% vs 16.9% respectively, *Figure 2H* and *Table 4*). We concluded that this intersectional strategy is suitable for the specific manipulation of serotonergic hindbrain neurons that project to the spinal dorsal horn, and that the majority of these traced cells are located in the LPGi.

**Table 4.** Cell counts for neurons labeled in the hindbrain from intersectional labeling experiments, from spinal cord injection of AAV2retro.flex.FLPo.mCherry and hindbrain injection of AAV9.FRT.eGFP (n=4 animals).
Total number of cells counted is given with the range of cells counted in parentheses.

| Fluorophore | Total | LPGi | medial | Other |
|---|---|---|---|---|
| eGFP only | 94 (19–29) | 18 (1–8) | 38 (6–16) | 38 (5–20) |
| eGFP +mCherry + | 558 (72–224) | 298 (50–96) | 156 (13–69) | 104 (6–59) |
| mCherry only | 153 (22–65) | 71 (11–23) | 55 (4–29) | 27 (3–13) |

## Lateral and midline serotonergic hindbrain neurons display anatomical differences in their spinal cord innervation

Previous anterograde tracing studies have suggested that within the RVM, the innervation of the superficial dorsal horn originates from the lateral hindbrain, whereas the deeper dorsal horn is innervated by the medial hindbrain (*Gautier et al., 2017*). The preferential labeling of lateral serotonergic neurons using our intersectional strategy provided us an opportunity to compare the projection patterns of different serotonergic neurons of the hindbrain.

To assess the serotonergic innervation of the spinal cord that originates from the medial RVM we injected AAVs containing Cre-dependent reporters into the NRM of Tph2-Cre mice (Coordinates from Bregma = –6, 0, 5.9; *Figure 3A*) and were able to label midline serotonergic neurons with eGFP without transduction of the LPGi neurons (*Figure 3B*). In addition to the NRM, neighboring midline regions were also labeled, such as the raphe obscuris (ROb) and raphe pallidus (RPa) (*Dahlström and Fuxe, 1964*). Most eGFP-labeled neurons in the injection site contained a detectable level of TPH2 (80.5%), indicating this strategy can be used to label the serotonergic midline neurons of the hindbrain (*Figure 3C*).

We used tdTomato-expressing vectors to visualize the axon termination pattern of descending serotonergic projections in the spinal cord (*Figure 3D and F*). For preferential labeling of the LPGi descending neurons, the lumbar spinal cord was injected with AAV2retro vectors containing a Cre-dependent optimized Flippase (AAV2retro.flex.FLPo.BFP), and 1 week later the hindbrain was injected with AAVs containing flippase-dependent tdTomato (AAV9.FRT.tdTOM). The stronger fluorescence of tdTomato enabled more sensitive labeling of the distal axons of descending projection neurons. The intersectional approach revealed that the densest labelling of axon terminals was present in the superficial laminae of the dorsal horn (*Figure 3D, E and H*). When the spinal cords were immunostained for CGRP and PKCγ (to delineate laminae I-IIo and IIi boundaries, respectively), most of the labeled axons were seen within the dorsal CGRP plexus, indicating that these neurons innervate laminae I-IIo of the spinal cord (*Figure 3—figure supplement 1A*). Surprisingly, projections retrogradely labeled from the left lumbar spinal cord contained axon collaterals that projected to both ipsilateral and contralateral dorsal horns, as well as to spinal segments caudal to the spinal injection site in the lumbar segment (*Figure 3E and H*). This indicates that these descending neurons have a wide-ranging axon termination pattern that extends over most of the spinal cord.

In contrast, axons from midline serotonergic neurons (including NRM, ROb, and RPa) directly labeled with tdTomato (from injection of AAVs containing a Cre-dependent tdTOM [AAV8.flex.tdTOM] into the NRM [–6, 0, 5.9]) were rarely present in the superficial laminae of the dorsal horn, but they densely innervated the ventral horn and were present in the deep dorsal horn (*Figure 3G, I*). This was confirmed by revealing the laminar boundaries with CGRP and PKCγ staining, which indicated that the axons of these neurons was mostly present ventral to the lamina IIi-III border (*Figure 3—figure supplement 1B*) In both labeling experiments, most of the axon terminals in the spinal cord contained 5-HT, although much of the spinal 5-HT was not colocalized with tdTomato by either labeling approach (*Figure 3H, I*). Together this indicates that the lateral and midline serotonergic neurons in the RVM project to distinct regions of the spinal cord, in agreement with *Gautier et al., 2017*.

## Midline and lateral serotonergic hindbrain neurons have similar active and passive membrane properties

Although much is known about the electrophysiological properties of serotonergic hindbrain neurons, to date there has not been a direct comparison of the membrane properties of those located in the NRM or LPGi. To test whether these distinct populations have similar biophysical membrane properties, Tph2-Cre neurons were labeled with tdTomato by hindbrain injection of AAV9.flex.tdTom and targeted for whole-cell recording.

In contrast to their anatomical differences (such as cell soma location and spinal axon termination pattern), Tph2-Cre neurons in the LPGi and NRM had similar active membrane properties and neuronal excitability (*Figure 4A and B*). In response to depolarizing current steps, these two groups of neurons discharged action potentials that were similar in terms of their thresholds, kinetics, and firing rates (*Figure 4A, B and D–G*). Cells were labeled with Biocytin during recording, and these groups were distinguished based on the location of biocytin-labeled cell bodies in the hindbrain slice after recording and tissue processing (*Figure 4C*). Similar to previous reports on the electrophysiological

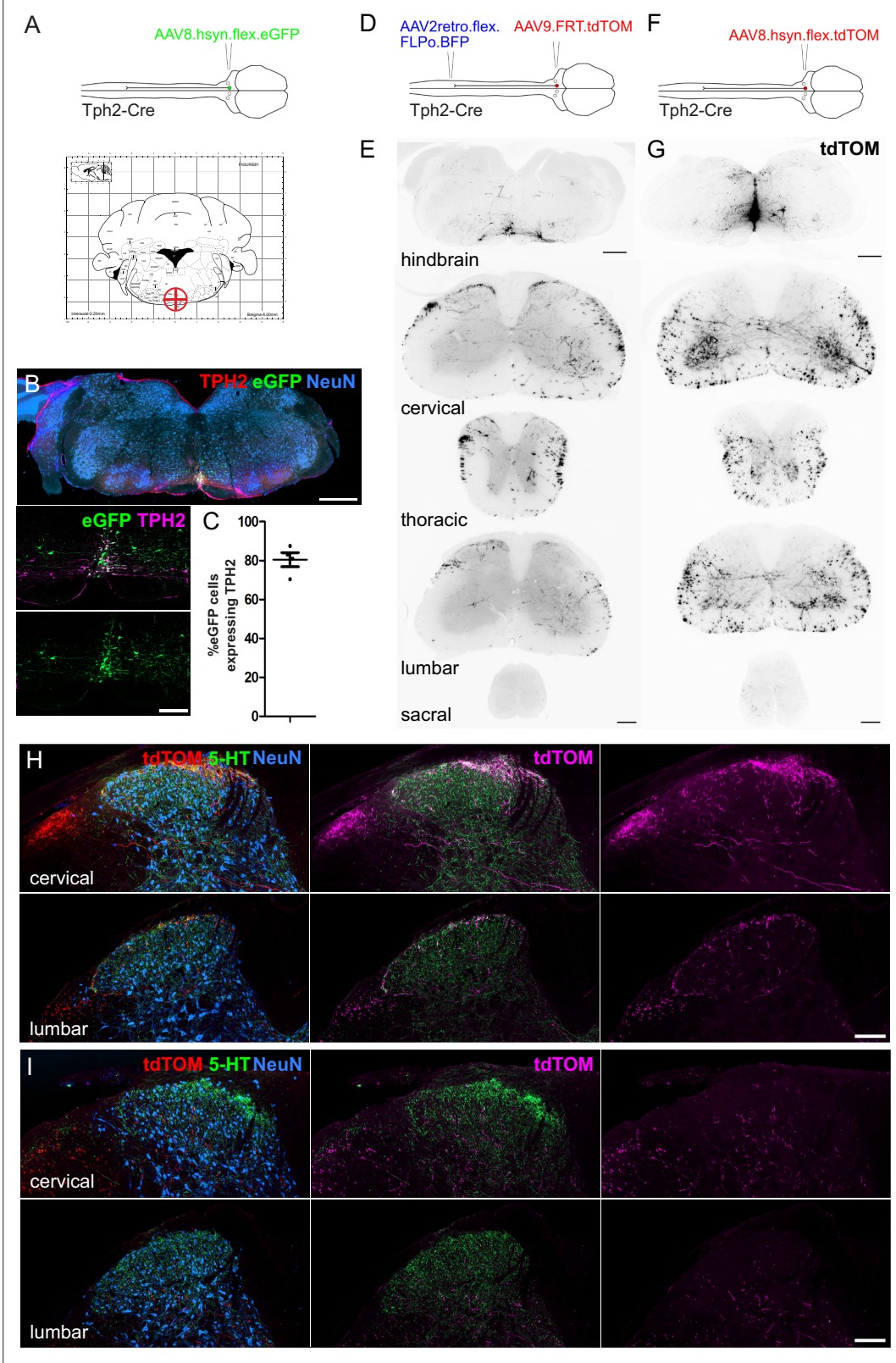

**Figure 3.** Spinal cord regions innervated by serotonergic hindbrain neurons. (**A**). Injection scheme for labeling medially located serotonergic neurons of the hindbrain with AAVs. Stereotaxic injection coordinates for the hindbrain injection of AAVs to label midline neurons without transducing LPGi serotonergic neurons (–6, 0, 5.9). (**B**). Representative brain injection site from the hindbrain of a Tph2-Cre mouse that received a single 300 nl

*Figure 3 continued on next page*

*Figure 3 continued*

injection of AAV8.hSyn.flex.eGFP (Scale bar = 500 μm). A higher magnification image of the injection site is also shown (scale bar = 100 μm). (**C**). Quantification of the proportion of eGFP-labeled neurons in the injection site that were immunoreactive for TPH2. (**D**). Intersectional strategy to preferentially label the spinal axon terminals of descending serotonergic neurons that originate in the LPGi. (**E**). Representative images of the hindbrain injection site and the axon termination pattern in the spinal cord of a Tph2-Cre animal that received the injections depicted in D (scale bars = 500 μm and 100 μm for hindbrain and spinal cord sections respectively). (**F**). Injection scheme for labeling medially located serotonergic neurons with tdTOM. (**G**). Representative images of the hindbrain injection site and spinal cord axon termination pattern of a Tph2-Cre animal that received a single 300 nl injection of AAV8.hSyn.flex.tdTOM depicted in F (scale bars = 500 μm and 100 μm for hindbrain and spinal cord sections respectively). (**H**). Higher magnification images of the cervical and lumbar dorsal horns of a Tph2-Cre animal that received the injections depicted in **D** (scale bar = 100 μm). (**I**). Representative images from cervical and lumbar spinal cord segments of animals that received an injection with AAV8.hSyn.flex.tdTOM into the NRM (scale bar = 100 μm).

The online version of this article includes the following figure supplement(s) for figure 3:

**Figure supplement 1.** Laminar location of axons originating from descending LPGi and midline serotonergic neurons of the hindbrain.

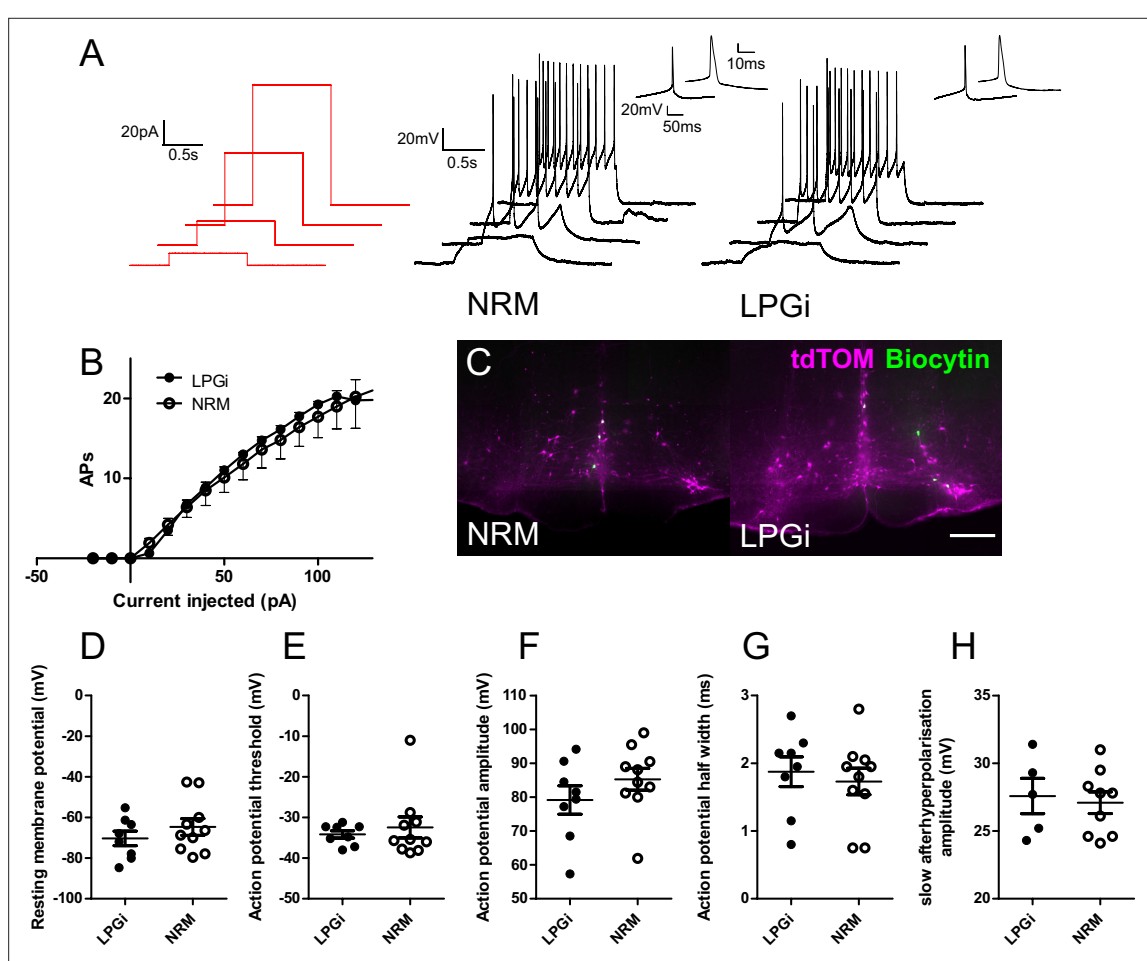

**Figure 4.** Electrophysiological characterization of serotonergic neurons of the NRM and the LPGi. (**A**). Current step protocol (red) and resulting AP firing (black) from representative serotonergic NRM and LPGi neurons recorded in current clamp. (**B**). Current/frequency plot for the AP firing frequency of NRM and LPGi neurons resulting from increasing 1 s depolarizing current injections. (**C**). Representative images of biocytin filled neurons within the NRM and LPGi of hindbrain slices revealed following recording (scale bar = 200 μm). (**D – H**) comparison of active and passive membrane properties between NRM and LPGi serotonergic neurons. (**D – G**). n=10 NRM (from 4 animals) n=8 LPGi (from 3 animals). (**H**). n=9 NRM n=5 LPGi.

**Table 5.** Comparison of active and passive membrane properties between serotonergic neurons in the NRM and the LPGi.

Data are shown as mean ± SEM (n=10 cells from 4 animals NRM, n=8 cells from 3 animals LPGi).

| Parameter | NRM | LPGi |
| --- | --- | --- |
| Membrane resistance (MΩ) | 1643±240.1 | 1857±316.9 |
| Resting membrane potential (mV) | −64.69±4.145 | −70.32±3.592 |
| Rheobase current (pA) | 18.75±2.236 | 15.00±2.266 |
| Action potential threshold (mV) | −32.45±2.597 | −34.18±0.8890 |
| Action potential amplitude (mV) | 85.26±3.230 | 79.14±4.194 |
| Action potential half width (ms) | 1.730±0.1960 | 1.875±0.2198 |
| Slow Afterhyperpolarisation (with AHP / total) | 9/10 | 5/8 |
| Slow Afterhyperpolarisation Amplitude (mV) | 27.09±0.7983 | 27.58±1.304 |

properties of serotonergic RVM neurons, both groups generally had broad action potentials (measured as width at half maximal) and frequently exhibited a slow afterhyperpolarization (9/10 NRM neurons, 5/8 LPGi neurons), which was similar in amplitude between groups (*Figure 4A, E and H*; *Zhang et al., 2006*). A summary of these electrophysiological measurements can be found in *Table 5*.

## Acute activation of descending serotonergic neurons in the LPGi decreases thermal sensitivity

Descending serotonergic projection neurons can both decrease or increase pain sensitivity, and optogenetic activation of serotonergic neurons of the NRM produces long-term hypersensitivity to both mechanical and thermal sensitivity (*Cai et al., 2014*; *Wei et al., 2010*). However, intrathecal injection of serotonin can produce acute antinociception (*Wang, 1977*), and currently little is known of the effect of preferentially activating descending serotonergic neurons in the LPGi.

To clarify whether the acute activation of these serotonergic neurons is pro- or antinociceptive, we used the intersectional strategy to express the excitatory DREADD hM3D(q) in descending serotonergic neurons of the LPGi (*Figure 5A*). Although many neurons of the hindbrain were labeled with hM3Dq-mCherry and virtually all mCherry-expressing axon terminals in the lumbar spinal cord contained detectable 5-HT, this only corresponded to a minority of the serotonergic innervation in this area (*Figure 5B and C*). After injecting CNO hM3Dq-mCherry-expressing neurons increased their activity, as evidenced by an increased expression of c-Fos in their nuclei relative to vehicle injected controls (*Figure 5—figure supplement 1*).

Despite sparse terminal labeling in the spinal cord, withdrawal latencies to thermal stimuli were increased following CNO injection (*Figure 5D*). Animals that had received CNO injections exhibited significantly longer withdrawal thresholds to infrared heat stimulation than vehicle injected controls (Repeated measures one-way ANOVA, $F_{(3, 9)}=10.55$, $p<0.0001$). Similarly, response latencies to the cold plantar assay were also prolonged following CNO injection (Repeated measures one-way ANOVA, $F_{(3, 9)}=7.309$, $p=0.0012$). In contrast, tactile sensitivity to punctate mechanical stimulation with von Frey filaments was unaltered (Repeated measures one-way ANOVA, $F_{(3, 9)}=1.402$, $p=0.2666$) (*Figure 6E*). To exclude the possibility that any observed alteration in sensory stimulus-induced behavior was due to impaired motor control or sedation, we measured the latency to fall from an accelerating rotarod before and after injection of CNO or vehicle. Latency to fall was comparable between animals before and after injection indicating no deficits in sensorimotor function or the absence of sedative effects (repeated measures one-way ANOVA, $F_{(3, 5)}=0.354$, $p=0.787$; *Figure 5D*) Together, these data indicate that acute activation of descending serotonergic neurons reduces sensitivity to thermal stimuli, but does not strongly influence mechanical sensitivity.

The presence of axon collaterals on the contralateral side of the spinal cord raises the possibility that these could also mediate a similar effect to those on the ipsilateral side. Therefore, we also tested the contralateral paw using the same sensory tests (*Figure 5—figure supplement 2*). We saw a significant increase in withdrawal latencies to heat stimulation in the Hargreaves assay (Repeated measures

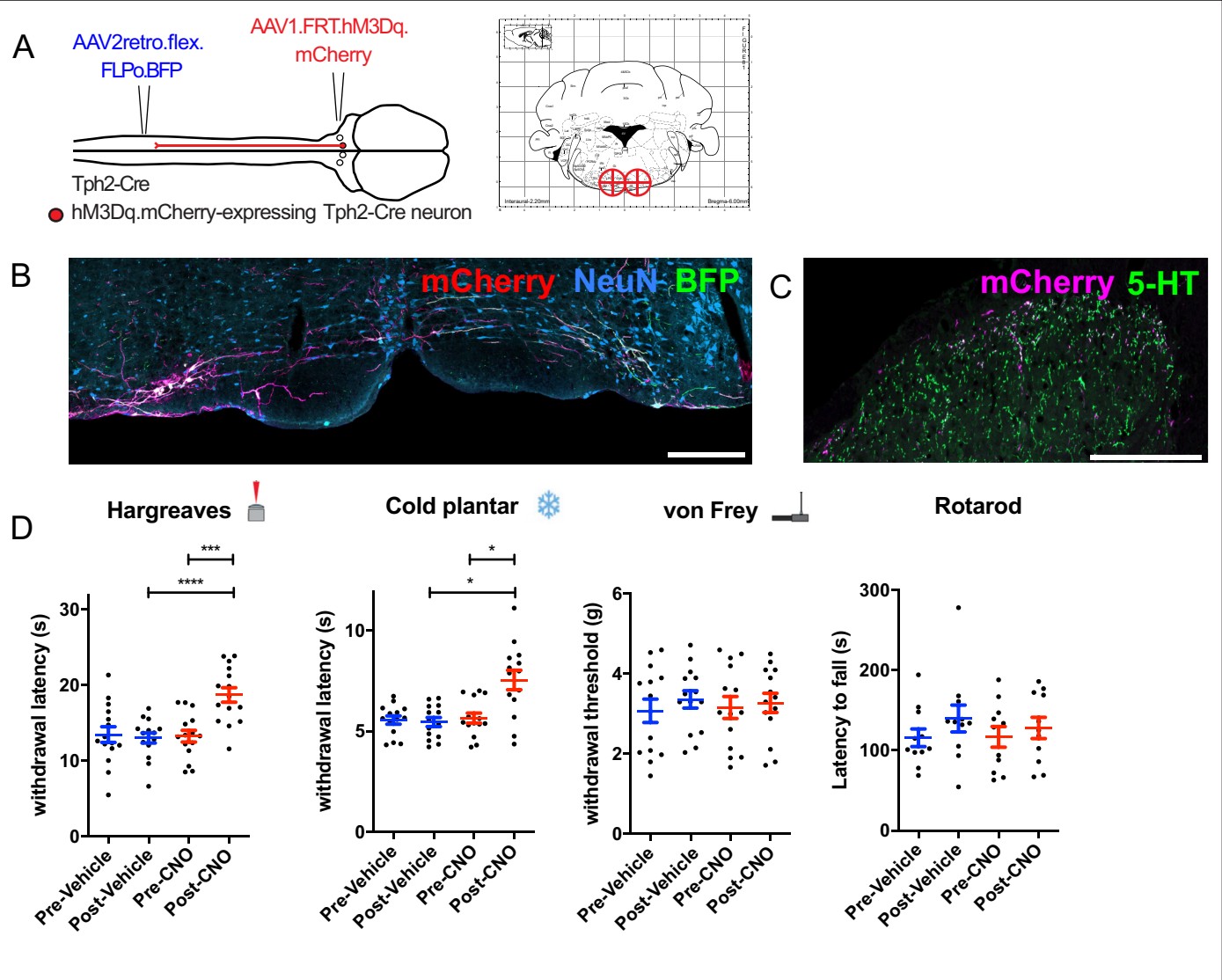

**Figure 5.** Chemogenetic activation of descending serotonergic LPGi neurons. (**A**). Injection scheme for expressing the excitatory DREADD hM3Dq in spinally projecting Tph2-Cre neurons. Brain injection coordinates according to the mouse brain atlas (−6,+/-0.5, 5.9 from bregma). (**B**). Example of the injection site from the hindbrain of a mouse that received a spinal dorsal horn injection of AAV2retro.flex.FLPo.BFP followed by a bilateral hindbrain injection of AAV1.FRT.hM3Dq.mCherry 1 week later (scale bar = 200 μm). (**C**). Example of the ipsilateral spinal dorsal horn form an animal that had received the injections indicated in **A**. Note that most 5-HT-containing terminals are not labeled, but the majority of labeled terminals contain a detectable level of 5-HT (scale bar = 200 μm). (**D**). Sensory tests of the ipsilateral hindpaw: Hargreaves plantar assay; repeated measures one-way ANOVA, ($F_{(3, 42)}=16.93$, $p<0.0001$), post hoc tests with Bonferroni's correction detected differences between post-vehicle and post CNO, as well as pre-CNO and post-CNO (adjusted p values are $p=0.007$ and $p<0.0001$ respectively). Cold plantar assay; repeated measures one-way ANOVA, ($F_{(3, 39)}=12.41$, $p<0.0001$) post hoc tests with Bonferroni's correction detected differences between post-vehicle and post CNO, as well as pre-CNO and post-CNO (adjusted p values are $p=0.0122$ and $p=0.0103$, respectively). von Frey test; repeated measures one-way ANOVA, ($F_{(3, 39)}=0.5013$ $p=0.6836$). Rotarod test for sensorimotor coordination/sedation; repeated measures one-way ANOVA, ($F_{(3, 30)}=0.8684$, $p=0.4683$). Significance: *$p<0.05$, **$p<0.01$, ***$p<0.001$.

The online version of this article includes the following figure supplement(s) for figure 5:

**Figure supplement 1.** Chemogenetic activation of hM3Dq-labeled neurons.

**Figure supplement 2.** Altered sensitivity of the contralateral paw following LPGi activation.

**Figure supplement 3.** Both male and female mice show alterations in thermal thresholds following chemogenetic activation of LPGi serotonergic neurons.

**Figure supplement 4.** CNO does not alter response latencies, thresholds or sensorimotor coordination in the absence of hM3Dq.

**Figure supplement 5.** Proportion of AAV2retro-traced neurons labeled with hM3Dq-mCherry for behavioral experiments.

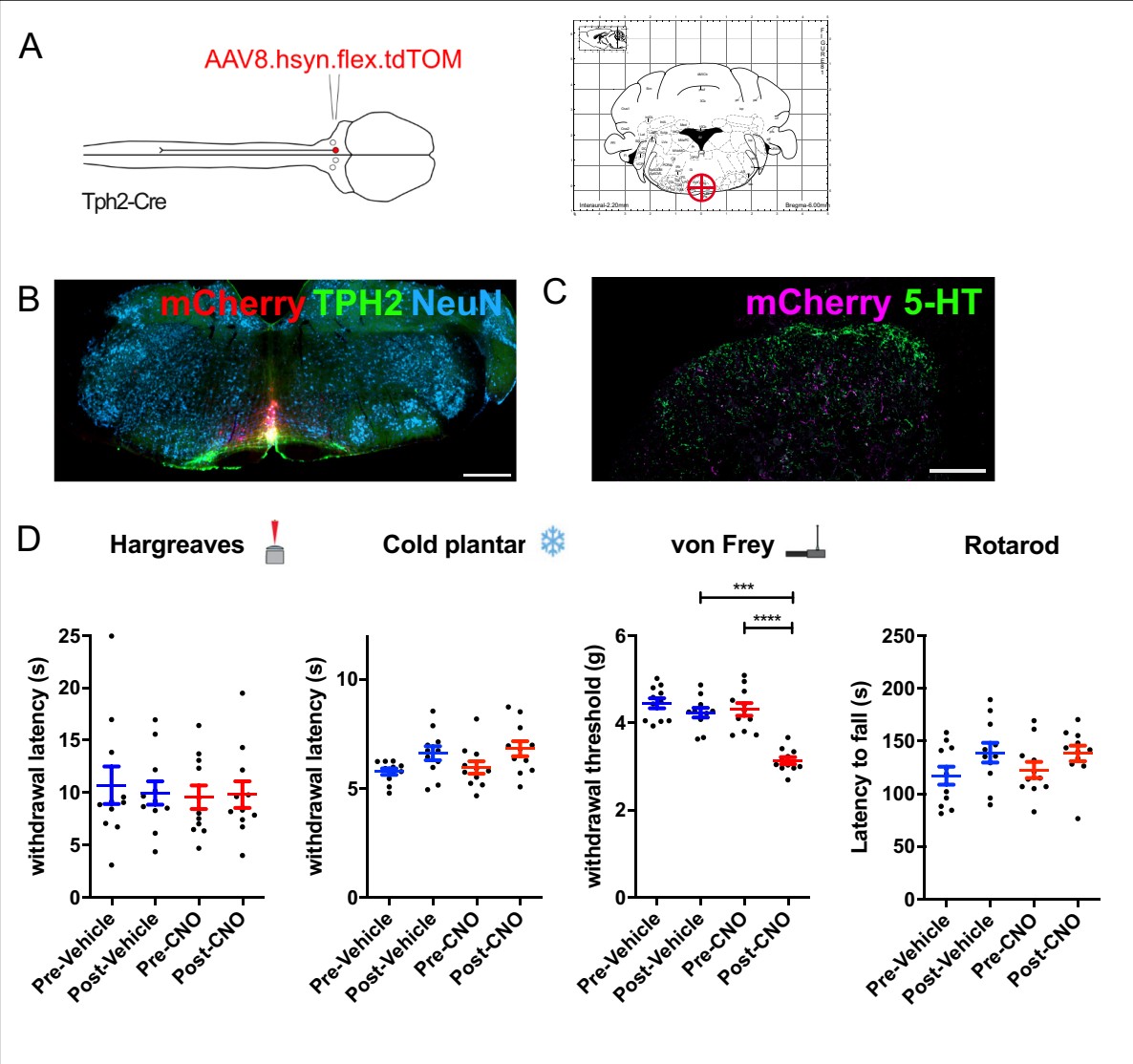

**Figure 6.** Chemogenetic activation of medial serotonergic hindbrain neurons. (**A**). Injection scheme for chemogenetic activation of medial serotonergic hindbrain neurons. Stereotaxic coordinates used for activating midline serotonergic neurons of Tph2-Cre animals with hM3D(q)-containing AAVs (–6, 0, 5.9). (**B**). Example of an injection site from an experiment to activate midline serotonergic neurons with hM3D(q) (scale bar = 500 μm). (**C**). Lumbar spinal cord from the injection site shown in **B**. showing mCherry-expressing terminals located ventral to the dense 5-HT innervation of the superficial dorsal horn (scale bar = 200 μm). (**D**). Sensory and sensorimotor coordination assays of animals following CNO or vehicle injections. Repeated one-way ANOVA, followed by Bonferroni's post-hoc tests for von Frey (F(3, 6)=13.84, p<0.0001) show significant decreases between post-vehicle and post-CNO injections (*P*=0.0005) as well as pre-CNO and post-CNO (p<0.0001). Significance: *p<0.05, **p<0.01, **p<0.001.

The online version of this article includes the following figure supplement(s) for figure 6:

**Figure supplement 1.** Injection sites for chemogenetic activation of medial serotonergic neurons of the hindbrain.

**Figure supplement 2.** Both male and female animals show increase in mechanical hypersensivity during chemogenetic activation of medial serotonergic hindbrain neurons.

**Figure supplement 3.** Proportion of neurons labeled in chemogenetic experiments that express TPH2.

one-way ANOVA, F(3, 14)=7.757, p=0.0003; *Figure 5—figure supplement 3*), suggesting that the activation of these neurons has some influence on the sensitivity of the contralateral paw, presumably via axon collaterals present in the contralateral dorsal horn.

Sex differences are often reported in the context of sensory neurobiology, and serotonin specifically has been implicated in mediating such differences (*Butkevich et al., 2007*; *Kaur et al., 2018*; *Mogil, 2012*). To see if similar differences are observed in our experiment, we divided the group into

male and female animals to determine whether both sexes responded in a similar manner. We find that both male and female mice show reduced sensitivity to thermal stimuli, and both show unaltered mechanical sensitivity (*Figure 5—figure supplement 3*).

In addition, CNO has been reported to mediate effects independently of hM3Dq expression (*Gomez et al., 2017*). To exclude that the effects seen in our assays were caused by such off-target effects, we performed the same sensory and motor tests on animals that did not express hM3Dq. In our hands, the injection of CNO did not produce any noticeable change in any of the behavioral tests in the absence of hM3Dq expression (*Figure 5—figure supplement 4*).

When we examined the hindbrain tissues from animals used in the LPGi activation experiments, we quantified the proportion of retrogradely labeled and intersectionally labeled neurons. On average, 67% of BFP-labeled neurons were also labeled with hM3Dq-mCherry, and very few (1.85%) only expressed mCherry (*Figure 5—figure supplement 5*). This shows that the intersectional strategy employed for chemogenetic activation of descending serotonergic neurons is similarly specific as our previous labeling experiments (*Figure 2F, G and H*).

## Acute activation of serotonergic neurons in the medial RVM increases mechanical sensitivity

Activation of serotonergic neurons has also been associated with increased sensitivity to tactile and thermal stimuli and is thought to underlie certain forms of neuropathic pain (*Cai et al., 2014*; *Suzuki et al., 2002*). The differences between our findings and those of *Cai et al., 2014* could be due to either of the mode/strength of neuronal activation (chemogenetic vs optogenetic) or the anatomical location of the activated neurons (medial hindbrain vs lateral hindbrain). To test this directly, we used the same chemogenetic receptor as our previous experiments to activate midline serotonergic neurons (*Figure 6A*). Using the same strategy as the anatomical tracing experiment (*Figure 3A*), we injected AAVs containing Cre-dependent hM3Dq.mCherry into the NRM. Similarly, we were able to limit transgene expression to the medial serotonergic neurons including the NRM, the ROb and RPa, but without spread to the LPGi (*Figure 6B* and *Figure 6—figure supplement 1*). This resulted in the presence of mCherry-labeled terminals in the deep dorsal horn of the spinal cord, with limited expression in the dense plexus of 5-HT terminals in the superficial spinal laminae (*Figure 6C*).

In contrast to the activation of serotonergic neurons within the LPGi, the activation of medial RVM neurons had no effect on thermal withdrawal latencies (Repeated measures one-way ANOVA, $F_{(3, 6)}=0.531$, p=0.667) or responses to cooling (Repeated measures one-way ANOVA, $F_{(3, 6)}=3.071$, p=0.054; *Figure 6D*). However, we observed a significant decrease in the withdrawal thresholds to punctate mechanical stimuli in the hindpaw (Repeated measures one-way ANOVA, $F_{(3, 6)}=13.84$, p=0.787; *Figure 6D*). Again, neither CNO nor vehicle injection impaired sensorimotor coordination or induced sedation, using latency to fall from an accelerating rotarod as a readout measure (Repeated measures one-way ANOVA, $F_{(3, 6)}=1.791$, p=0.185; *Figure 6D*).These data indicate that, in contrast to serotonergic neurons of the LPGi, activating medial serotonergic hindbrain neurons increases mechanical sensitivity without influencing responses to thermal stimuli.

To test for potential sex differences in mechanical hypersensitivity, we again divided our group into males and females and compared the withdrawal thresholds/latencies post-vehicle injection to post-CNO injection. We see that both male and female animals exhibit similar increases in mechanical sensitivity after CNO injection (*Figure 6—figure supplement 2*), suggesting that this effect is not sex specific. The withdrawal latencies to heat and cold were unchanged and the latency to fall from the accelerating rotarod were similarly unchanged in both male and female animals.

Additionally, when we inspected the injection sites in the brain stem from animals used in these behavioral experiments, we found that a similar proportion of neurons expressed TPH2 (85.1%, *Figure 6—figure supplement 3*) as compared to our previous labeling experiments (*Figure 3C*). Therefore, we concluded that the chemogenetic activation is largely restricted to serotonergic neurons of the medial hindbrain in these experiments.

## Discussion

### Anatomical organization of descending serotonergic innervation of the lumbar spinal cord

The anatomical locations of serotonergic nuclei in the brainstem have been known since the early studies of Dahlstrom and Fuxe (*Dahlström and Fuxe, 1964*). These areas have later been shown to contain spinally-projecting neurons and together provide the spinal cord with serotonin (*Kwiat and Basbaum, 1992*). Similarly, we were also able to label the same regions from the spinal cord with CTb tracing (*Figure 1—figure supplement 2*). However, we found that most of the descending serotonergic neurons transducible with AAV2retro serotype vectors from the mouse spinal cord are located in the LPGi, similar to what has been previously demonstrated (*Wang et al., 2018*). We used this selectivity to devise a strategy to preferentially label midline or lateral serotonergic neurons and demonstrate that, in agreement with anterograde tracing studies, the innervation of the superficial laminae of the dorsal horn mainly originates from the LPGi and not the NRM (*Gautier et al., 2017*). Unlike (*Gautier et al., 2017*), we found many axon terminals present within the ventral horn as well as the deep dorsal horn. Potentially this reflects a species difference (rat versus mouse), or the labeling of other midline serotonergic regions beyond the NRM, such as the ROb or the RPa in the present study (*Hennessy et al., 2017*; *Okaty et al., 2019*). However, inclusion of these midline serotonergic neurons within the injection site demonstrates that they do not project to the superficial dorsal laminae. Therefore, it is likely that these pathways influence anatomically distinct neuron populations of the spinal cord, and likely play different functional roles.

The spinal dorsal horn is organized in a laminar structure that is closely linked to its function, such that thermal and nociceptive information is generally received and processed in the superficial laminae, whereas non-nociceptive mechanical and proprioceptive information is processed in deeper spinal laminae (*Todd, 2010*). This laminar organization can be seen in terms of gene expression (*Häring et al., 2018*), the termination zones of different primary sensory neurons (*Cavanaugh et al., 2009*; *Li et al., 2011*), and the restricted laminar location of neuron populations (*Das Gupta et al., 2021*). Therefore, the laminar location of serotonin release will mostly influence different populations of neurons within these regions, which play diverse functional roles (*Albisetti et al., 2019*; *Duan et al., 2014*; *Foster et al., 2015*; *Gatto et al., 2021*).

### Antinociceptive function of descending LPGi serotonergic neurons in spinal circuits

Acute activation of spinally projecting serotonergic neurons using excitatory DREADDs was able to reduce sensitivity to thermal tests (*Figure 5D*). In support of this idea, the axons of these projections are densest in the superficial laminae of the spinal cord (*Figure 3E and H*), where temperature-related information is conveyed by TRPV1- and TRPM8-expressing sensory neurons (*Cavanaugh et al., 2011*; *Cavanaugh et al., 2009*; *Dhaka et al., 2008*). The presence of axons from lateral serotonergic neurons within the region that receives, and processes thermal information is consistent with the observation that activating these neurons affects thermal sensitivity. Furthermore, 5-HT can inhibit C fiber-mediated input to the rat superficial dorsal horn and hyperpolarize many types of excitatory neurons in this area, in agreement with a reduction in neuronal activity that would inhibit the relay of temperature-related information (*Lu and Perl, 2007*).

### Pronociceptive function of medial serotonergic neurons in spinal circuits

Preclinical models of chronic pain can increase the activity of serotonergic hindbrain neurons, and TPH2 knockdown experiments can transiently reduce spinal 5-HT and hypersensitivity in neuropathic animals (*Suzuki et al., 2002*; *Wei et al., 2010*). In these neuropathic models, several changes in the physiology of the dorsal horn are widely reported (*Coull et al., 2003*; *Todd, 2015*; *West et al., 2015*). Together, these pathological changes may alter the way that 5-HT influences the processing of sensory information within the dorsal horn. In agreement, the spinal serotonin receptors engaged by endogenous pain control systems such as diffuse noxious inhibitory controls are altered during the development of chronic pain (*Bannister and Dickenson, 2016*; *Bannister et al., 2017*; *Bannister et al.,*

*2015*). Additionally, alterations could also occur within the serotonergic projection neurons, such as their activity and neurotransmitter content/concentration (*Tudeau et al., 2020*).

However, there are also data that point to a pro-nociceptive role for 5-HT at the spinal level in naïve animals. For example, it was shown that optogenetic stimulation of the NRM of TPH2-ChR2 animals could induce hypersensitivity that lasted several weeks (*Cai et al., 2014*). In our chemogenetic experiments, the activation of NRM neurons with CNO did produce an increase in mechanical sensitivity, but unlike (*Cai et al., 2014*) we did not see an alteration in thermal sensitivity or an effect that outlasted the neuron stimulation. These differences could be explained in part by the intensity and location of the stimulation used in experiments. In support of this idea, activation of the same neuron population with optogenetic or chemogenetic tools can produce distinct behaviors, likely due to differences in the strength or pattern of the neuronal activation (*Sharif et al., 2020*).

The activation of the midline serotonergic neurons influences multiple functionally different groups of neurons within the injection site, including the NRM, ROb, and RPa (*Okaty et al., 2019*). These more caudal nuclei are known to affect the motor system and respiratory functions (*Brust et al., 2014*; *Hennessy et al., 2017*; *Iceman et al., 2013*). Within the sensory assays tested, there was a rather selective change in mechanical responses and unaltered thermal responses, indicating that the effect was modality selective and unlikely due to motor effects alone. However, it cannot be excluded that serotonergic neurons in these distinct midline regions contribute to the reduced mechanical thresholds, either indirectly or by direct altering motor neuron activity in the ventral horn (*Depuy et al., 2011*; *Hennessy et al., 2017*; *Kawashima, 2018*). Further studies of sensory function utilizing intersectional strategies to precisely capture and manipulate the midline populations will help to validate the present findings (*Okaty et al., 2019*).

Together the results of the chemogenetic experiments indicate that the same transmitter released into different spinal laminae can produce opposing effects on spinal nociception. Potential explanations for these findings include the 5-HTR subtypes that are engaged, the spinal neuron populations that are modulated, or a combination of both. Some 5-HTRs are known to have an excitatory influence when activated, whereas others have an inhibitory effect upon ligand binding (*Abe et al., 2009*; *Bardoni, 2019*; *Lu and Perl, 2007*). The activation of excitatory neurons and the inhibition of inhibitory neurons in the deep dorsal horn, corresponding to the termination zone of the medial serotonergic neurons, are known increase mechanical sensitivity (*Duan et al., 2014*; *Foster et al., 2015*; *Peirs et al., 2021*; *Peirs et al., 2015*). Conversely, the activation of inhibitory neurons in the superficial dorsal horn, the region innervated by the lateral serotonergic neurons, and the inhibition of nociceptors through presynaptic inhibition of their central terminals could promote thermal hyposensitivity (*Jeong et al., 2012*; *Lu and Perl, 2007*; *Xie et al., 2012*). Further studies are required to elucidate precisely which spinal 5-HT receptors and neuronal elements are mediating these opposing sensory phenomena.

Given the previously reported heterogeneity in biophysical properties of serotonergic neurons (*Okaty et al., 2019*), we found surprisingly little difference in the passive and active membrane properties between NRM and LPGi serotonergic neurons (*Table 5* and *Figure 4*). Additionally, none of the recorded neurons displayed spontaneous activity, which has been reported previously for serotonergic projection neurons of the hindbrain (*Zhang et al., 2006*). This may be attributable to differences in recording conditions, such as recording temperature, animal species, and recording solution composition. Additionally, these groups of neurons may exhibit differences in electophysiological properties, such as chemosensitivity, spontaneous activity, and responses to nociceptive stimuli, that are only detected in in vivo preparations or under recording conditions that differ from the present study (*Gao and Mason, 2000*; *Gao and Mason, 2001*; *Hennessy et al., 2017*; *Mason et al., 2007*; *Potrebic et al., 1994*; *Richerson et al., 2001*).

## Limitations of transsynaptic rabies tracing of serotonergic RVM neurons

Transsynaptic rabies tracing has been widely used in recent years to identify presynaptic inputs to genetically defined neuronal populations and has become a frequently used tool for the study of neuronal circuits. Functional studies are limited due to the toxicity of first generation ΔG rabies viruses. However, other limitations of this technique are known, such as the resistance of some neuron types to infection by modified rabies viruses (*Albisetti et al., 2017*). It is possible that some types of

hindbrain neurons are completely resistant to infection with modified rabies viruses, although this is not the case for serotonergic neurons, as they can be infected via their axon terminals (*Figure 1— figure supplement 4A–C*). We show that although descending TPH2-expressing neurons are not resistant to infection from G-protein-deficient rabies viruses, they are rarely traced transsynaptically from dorsal horn neurons (*Figure 1—figure supplement 4*). The lack of transsynaptic labeling from the dorsal horn is possibly due to the synaptic organization of serotonergic axons at this site, since serotonin is known to mainly act via volume transmission (*Ridet et al., 1993*). Since similar numbers of hindbrain neurons were labeled using direct rabies infection and transsynaptic rabies labeling, this could indicate an increased labeling efficiency of other neurons with the latter approach. Potentially this could be related to the synaptic density from these neurons to the starter population. However, both direct and transsynaptic rabies labeling were far less sensitive than the CTb or AAV2retro tracing, making interpretation of these findings complicated due to the comparatively small sample sizes.

Most serotonergic axons in the rat dorsal horn do not form synaptic contacts (*Maxwell et al., 1983*; *Ridet et al., 1993*). However, some 5-HT-containing axonal boutons do form synapses within the dorsal horn, and these are observed as symmetrical synapses (*Polgár et al., 2002*). Since projection neurons are approximately 5% of all lamina I neurons (*Cameron et al., 2015*), the transsynaptic tracing from these neurons may not be detected in our experiments. Additionally, transsynaptic tracing from a primary afferent starter population was used to label serotonergic hindbrain neurons, suggesting axo-axonic synapses between descending serotonergic neurons and their central terminals (*Zhang et al., 2015*). In the present study, the number of primary afferent neurons transduced from intraspinal injection of helper virus is likely small, since the AAV8 serotype and the inclusion of the hSyn promoter were used to reduce the transduction of sensory neurons (*Haenraets et al., 2018*). This is consistent with the low number of mCherry-expressing cells found in the DRG taken from these animals. Taken together, these may explain why so few serotonergic hindbrain neurons were traced transsynaptically from the spinal dorsal horn (*François et al., 2017*; *Liu et al., 2019*).

## Preferential transduction of LPGi serotonergic neurons with AAV2retro serotype vectors

The AAV2retro serotype has been developed for the retrograde delivery of genetic material to projection neurons via their axon terminals in a target region (*Tervo et al., 2016*). Many projection neurons are found to display some resistance to transduction with these tools, for example dopaminergic neurons that project from the substantia nigra to the striatum and descending noradrenergic neurons of the locus coeruleus to the spinal cord (*Ganley et al., 2021*; *Tervo et al., 2016*). Like others, we also find that midline neurons including the NRM serotonergic neurons are less sensitive to AAV2retro-mediated transduction (*Wang et al., 2018*). Surprisingly, we observed that the serotonergic neurons of the lateral hindbrain do not display such resistance (*Figure 1*). This would suggest that these neurons are different in some property that allows the AAV2retro to enter their terminals or enhances the retrograde transport of the viral payload. The increased retrograde labeling efficiency of AAV2retro is not fully understood, although enhanced entry into axon terminals by increased spread of the vector in the injection site, entry into neurons via novel cell surface receptor interactions, and vesicular trafficking have been suggested (*Tervo et al., 2016*). AAV2retro-resistant neurons commonly use monoamines as neurotransmitters, suggesting that they likely share common features protecting them from transduction by these vectors.

Primarily, our data demonstrate that serotonergic neurons of the lateral hindbrain are antinociceptive when activated. These neurons project their axons to the superficial laminae of the dorsal horn and can be targeted using AAV2retro vectors. In contrast, the medial serotonergic neurons are largely resistant to AAV2retro-mediated transduction, produce mechanical hypersensitivity upon acute activation, and project their axons to the deeper dorsal horn. Secondly, we highlight some of the limitations and challenges associated with AAV2retro-mediated and rabies-virus-based transsynaptic tracing from the spinal dorsal horn. These limitations are likely to have broader implications for transsynaptic rabies-based circuit tracing and the targeting of other projections that use biogenic amines, and possibly other transmitter systems.

**Table 6.** Transgenic mouse lines used in this study.

| Mouse line | Supplier/source | Reference |
|---|---|---|
| Tg(Tph2-Cre)RH35Gsat (Tph2-Cre) | The Jackson Laboratory | MGI: 5435520 |
| Tg(Hoxb8-cre)1403Uze (Hoxb8-Cre) | Pawel Pelczar | (*Witschi et al., 2010*) MGI:4881836 |
| Gt(ROSA)26Sor$^{tm1(Tva)Das}$ (ROSA26$^{TVA}$) | Dieter Sauer | (*Seidler et al., 2008*) MGI:3814188 |

## Materials and methods

### Animals

Mice of either sex aged between 6 and 12 weeks were used for experiments. Permission to perform these experiments was obtained from the Veterinäramt des Kantons Zürich (154/2018 and 063/2016). The various transgenic mouse lines used in this study are listed in *Table 6*. Tph2-Cre and Hoxb8-Cre mice are both BAC transgenic lines, and the ROSA$^{TVA}$ is a knockin reporter line.

### Surgeries

The dorsal horn of the lumbar spinal cord was injected in a similar manner to previous studies (*Ganley et al., 2021*; *Haenraets et al., 2018*). Briefly, anesthesia of mice was induced with 5% Isoflurane in an induction chamber. Mice were then transferred to a stereotaxic injection setup and anesthesia was maintained with 1–3% Isoflurane delivered through a face mask. Body temperature was maintained using a heated mat placed beneath the animal. Vitamin A cream was applied to the eyes to prevent corneal drying during the operation and buprenorphine (0.1–0.2 mg/kg) was injected subcutaneously prior to the operation. The back of the animal was shaved, and the skin was scrubbed with Betadine solution. Once dried, a midline incision was made above the vertebral column to expose the T13 vertebra, which was clamped using a pair of spinal adaptors to isolate movements from the animal's breathing. This vertebra was selected since it is directly above the lumbar L3 spinal cord, which corresponds to the spinal cord segment receiving innervation from the hindlimbs (*Haenraets et al., 2018*). A borehole was made in the center of the left-hand side of the clamped vertebra and viruses, and/ or 1% cholera toxin b (CTb) were injected into the dorsal horn at a depth of 300 µm below the spinal surface approximately 500 µm left of the central artery. For most injections, 3x300 nl virus solution was injected along the rostrocaudal extent of the spinal cord at an infusion rate of 50 nl/min. For transsynaptic tracing experiments, the rabies virus was injected 2x500 nl at either side of the T13 vertebra in the same region as the injection of helper virus, which was injected two weeks previously. For a list of the retrograde tracers and viruses used in this study, please refer to *Table 7*.

For injections into the hindbrain, animals were prepared for surgery in a similar manner to the spinal cord injections. The head was fixed in position using ear bars mounted on the stereotaxic frame (Kopf instruments). Coordinates were chosen based on the hindbrain location of the Tph2-Cre neurons retrogradely labeled from the spinal cord with reference to a mouse brain atlas (−6,+/-0.5, 5.9, 1 µl per injection site, infusion rate 50 nl/min). All coordinates are given as rostrocaudal, mediolateral, and dorsoventral (x, y, z) relative to Bregma. Injections into the midline NRM were made at coordinates –6, 0, 5.9 and a volume of 300 nl was chosen to limit the transgene expression to the medial serotonergic neurons. The movement of the frame was achieved using motorized axes controlled by a computer interface, which was also used to select the injection target (Neurostar). This same software was used to adjust the injection target for tilt and scaling, by adjusting the mouse brain atlas relative to four points on the surface of the skull (Bregma, Lambda, 2 mm to the right of the midline, and 2 mm to the left of the midline).

### General features of tissue preparation and immunohistochemistry

Animals were perfusion fixed with freshly depolymerized 4% paraformaldehyde (room temperature, dissolved in 0.1 M PB, adjusted to pH 7.4) following a brief rinse of the mouse circulatory system with 0.1 M PB. Nervous tissues were quickly dissected and post-fixed in the same fixative for two hours at 4 °C. After post-fixation, tissues were rinsed 3 times with 0.1 M PB and placed in 30% sucrose solution (w/v dissolved in 0.1 M PB) for 24–72 hr for cryoprotection. Tissues were rinsed with 0.1 M PB before

**Table 7.** AAVs, rabies viruses, and retrograde tracers used in the study.

| Virus/tracer name | Full name | Supplier/Source | Cat# |
|---|---|---|---|
| AAV2retro.eGFP | ssAAV-retro/2-CAG-EGFP-WPRE-SV40p(A) | Viral Vector facility UZH/ETHZ | V24-retro |
| AAV2retro.flex.eGFP | ssAAV-retro/2-shortCAG-dlox-EGFP(rev)-dlox-WPRE-SV40p(A) | Viral Vector facility UZH/ETHZ | V158-retro |
| AAV2retro.flex.tdTomato | ssAAV-retro/2-shortCAG-dlox-tdTomato(rev)-dlox-WPRE- hGHp(A) | Viral Vector facility UZH/ETHZ | V167-retro |
| AAV2retro.flex.FLPo.BFP | ssAAV-retro/2-hSyn1-chl-dlox-EBFP2_2 A_FLPo(rev)-dlox-WPRE-SV40p(A) | Viral Vector facility UZH/ETHZ | V175-retro |
| AAV2retro.flex.FLPo.mCherry | ssAAV-retro/2-hSyn1-chl-mCherry_2 A_FLPo-WPRE-SV40p(A) | Viral Vector facility UZH/ETHZ | V173-retro |
| AAV9.flex.ChR2-YFP | ssAAV-9/2-hEF1a-dlox-hChR2(H134R)_EYFP(rev)-dlox-WPRE-hGHp(A) | Viral Vector facility UZH/ETHZ | v214-9 |
| AAV8.FRT.tdTomato | ssAAV-8/2-hSyn1-dlox-tdTomato(rev)-dlox-WPRE-bGHp(A) | Viral Vector facility UZH/ETHZ | v284-8 |
| AAV9.FRT.eGFP | ssAAV-9/2-hSyn1-chl-dFRT-EGFP(rev)-dFRT-WPRE-hGHp(A) | Viral Vector facility UZH/ETHZ | V335-9 |
| AAV9.FRT.hM3D(q).mCherry | ssAAV-9-hSyn1-dFRT-hM3D(Gq)-mCherry | Viral Vector facility UZH/ETHZ | V189-9 |
| AAV9.FRT.hM4D(i).mCherry | ssAAV-9/2-hSyn1-dFRT-hM4D(Gi)_mCherry(rev)-dFRT-WPRE-hGHp(A) | Viral Vector facility UZH/ETHZ | V190-9 |
| AAV9.FRT.ChR2-YFP | ssAAV-9/2-hSyn1-dFRT-hM4D(Gi)_mCherry(rev)-dFRT-WPRE-hGHp(A) | Viral Vector facility UZH/ETHZ | V190-9 |
| AAV9/2.FRT.eGFP.TeTxLC | ssAAV-9/2-hSyn1.chl-dFRT-EGFP-2A-FLAG:TeTxLC(rev)-dFRT-WPRE-hGHp(A) | Viral Vector facility UZH/ETHZ | v450-9 |
| SAD pseudotyped rabies | SAD.RabiesDG-eGFP (SAD-G) | Karl-Klaus Conzelmann | N/A |
| EnvA pseudotyped rabies | SAD.RabiesDG-GFP (EnvA) | Karen Haenraets | N/A |
| CTb | Cholera Toxin b subunit | Sigma Aldrich | C9903-.5MG/ |

**Table 8.** Antibodies used in the study.

| Antibody | Host | Supplier/Source | Cat#/RRID | Dilution |
|---|---|---|---|---|
| GFP | Chicken | LifeTech | A10262/AB_2619988 | 1:1000 |
| TPH2 | Rabbit | Novus Biologicals | NB100-74555/AB_572263 | 1:1000 |
| Ctb | Goat | LIST biological laboratories inc. | #703/AB_2314252 | 1:1000 |
| mCherry | Goat | Sicgen | AB0081-200/AB_2333094 | 1:500 |
| tdTomato | Goat | Sicgen | AB8181−200/AB_2722750 | 1:500 |
| 5-HT | Rabbit | ImmunoStar | 20080/AB_572263 | 1:1000 |
| NeuN | Guinea pig | Synaptic systems | 266004/AB_2619988 | 1:1000 |
| Chicken-Alexa 488 | Donkey | Jackson ImmunoResearch | 703-546-155/AB_2340376 | 1:500 |
| Goat-Cy3 | Donkey | Jackson ImmunoResearch | 705-166-147/AB_2340413 | 1:500 |
| Goat-Alexa 488 | Donkey | Jackson ImmunoResearch | 795-546-147/ - | 1:500 |
| Guinea pig-Alexa 647 | Donkey | Jackson ImmunoResearch | 706-496-148/ - | 1:500 |
| Rabbit-Alexa 647 | Donkey | Jackson ImmunoResearch | 711-607-003/AB_2340626 | 1:500 |
| Rabbit-Cy3 | Donkey | Jackson ImmunoResearch | 711-165-152/AB_2307443 | 1:500 |

being embedded in NEG-50 mounting medium and were either cut at 60 µm on a sliding blade microtome (Hyrax KS 34, histocam AG) and stored as free-floating sections, or were cut at 30 µm using a cryostat (Hyrax 60, histocam AG) and were mounted directly onto microscope slides (Superfrost Plus, Thermoscientific).

Free floating sections were processed immediately for tissue staining or were stored in antifreeze medium 50 mM sodium phosphate buffer, 30% ethylene glycol, 15% glucose, and sodium azide (200 mg/L) at –20 °C until required. Antifreeze medium was removed by rinsing sections three times in 0.1 M PB before further processing. Sections were rinsed in 50% ethanol for 30 min at room temperature, followed by three rinses in PBS +8 g/L NaCl Alternatively, slides were prepared by mounting frozen sections directly onto superfrost slides. These were either used immediately for immunostaining or stored at –80 °C until further use. Before immunostaining, excess NEG50 freezing medium was removed by rinsing in 0.1 M PB for 1 hr at room temperature. A hydrophobic barrier was drawn around the sections using a fat pen, and immunoreactions were performed on the slides.

All primary antibodies and dilutions used in this study are included in *Table 8*. Primary antibodies were diluted in a PBS +8 g/L NaCl, 0.3% v/v Triton-X, and 10% v/v normal donkey serum. Secondary antibodies were diluted in a similar solution, but without the 10% normal donkey serum (*Table 8*). Sections or slides were incubated in primary antibodies for 24–72 hr at 4 °C, and these were revealed by incubation in secondary antibodies overnight at 4 °C. Following immunostaining, sections or slides were rinsed three times for 10 min each in PBS +8 g/L NaCl before being mounted in Dako anti-fade medium.

## Rabies virus tracing experiments

Rabies virus tracing experiments used either the SAD-G pseudotyped or EnvA pseudotyped RabΔG-GFP viruses, which are deficient in the SAD-glycoprotein required for transsynaptic spread and contain eGFP for identification of infected cells (*Wickersham et al., 2007a*; *Wickersham et al., 2007b*). Rabies viruses pseudotyped with the SAD-G glycoprotein can directly infect most, but not all, types of neurons (*Albisetti et al., 2017*; *Wickersham et al., 2007a*). The rabies viruses that are pseudotyped with the EnvA glycoprotein are only able to infect cells through binding to the TVA receptor. Therefore, the restricted expression of the TVA receptor to certain cell types allows the selective infection of those neurons with EnvA pseudotyped rabies viruses (*Wickersham et al.,*

*2007b*). For transsynaptic tracing experiments, the helper virus containing the rabies glycoprotein was injected two weeks prior to rabies virus injection to enable transsynaptic spread.

Animals were perfused 5 or 7 days after dorsal horn injection of rabies viruses. Hindbrain sections from rabies virus injected tissue were cut at 30 µm and mounted directly onto microscope slides. Tissues were immunostained for eGFP and TPH2 to determine the proportion of labeled hindbrain neurons that contained detectable TPH2.

## Image acquisition and analysis

For quantification of retrogradely labeled cells in the hindbrain, image stacks were acquired at 5 µm z-spacing using a Zeiss lsm 800 confocal microscope with Zen blue software. Confocal scans were made using 488, 561, and 640 nm lasers and the pinhole was set to 1 Airy Unit for reliable optical sectioning. Care was taken to acquire image stacks up to a depth where there was clear immunoreactivity of all antigens to avoid false negatives during quantification due to antibody penetration. Within each experiment, all acquisition settings were kept constant, and images were analyzed with FIJI using the cell counter plugin.

Alternatively, an automated cell quantification pipeline was designed in CellProfiler to count retrogradely labeled neurons in the RVM. We noticed that many large neurons within the RVM contained NeuN immunoreactivity that was present in more than one image in the image stacks (5 µm z-spacing). Automated counting of all images within each z-stack would lead to double counting a large proportion of cells present on multiple images within the image stack. However, if only one image was analyzed per image stack, numerous cells would be excluded from the analysis. Therefore, the acquired z-stacks were processed into either one to two orthogonal projections from two images to reduce the double counting of cells whilst ensuring the counting of most visible cells within each stack. One sample projection was then counted manually using the cell counter plugin of ImageJ, to serve as a guide in the development of the CellProfiler pipeline. All orthogonal projections were processed using the same CellProfiler pipeline with set parameters to ensure consistent data collection.

The images were loaded into this CellProfiler pipeline and the three channels (CTb, eGFP and NeuN) were separated from each other. NeuN- and CTb-stained neurons were identified as primary objects using the two-class Otsu method as a global thresholding strategy, whilst eGFP-containing objects were identified using the adaptive thresholding strategy (with an adaptive window of 50 pixels). NeuN stained cells were distinguished from the background based on their intensity, whereas the method used for detecting CTb and eGFP-positive cells was based on their shape. The NeuN objects were converted into a binary (black and white) image and the coexpression of eGFP or CTb was assessed. The object counts of eGFP/NeuN, CTb/NeuN, eGFP/CTb/NeuN, and NeuN-positive objects were then extracted and exported to a spreadsheet.

To determine whether serotonergic neurons were labeled using modified rabies virus tracing, eGFP-labeled neurons in the hindbrain were identified and immunostained against TPH2. These were scanned on a Zeiss 710 LSM confocal microscope, either as a short stack through the cell body, or as a single optical section through the center of the neuron. Scan settings were determined by the fluorescent intensity of the surrounding TPH2 immunoreactive regions in the NRM and lateral paragigantocellularis (LPGi) and were kept consistent for all experiments. Cell identities were catalogued and TPH2 immunoreactivity was determined for all sampled cells.

## Slice preparation and electrophysiology

Hindbrain slices were prepared from Tph2-Cre animals that had received a bilateral injection of AAV9. flex.tdTomato into the ventral hindbrain. Animals were aged 3–6 weeks at the time of injection and were prepared for electrophysiological recordings 1–2 weeks later. Slices were prepared in a similar manner to previous studies (*Pedersen et al., 2011*; *Zhang et al., 2006*). Animals were decapitated and the brain was rapidly dissected and placed in ice cold oxygenated dissection solution (containing in mM (65 NaCl, 105 sucrose, 2.5 KCl, 1.25 NaH$_2$PO$_4$, 25 NaHCO$_3$, 25 glucose, 0.5 CaCl$_2$, 7 MgCl$_2$). The hindbrain was isolated, glued to a block of 2% agarose and installed in a slicing chamber. Transverse slices of hindbrain were cut at 250 µm on a vibrating blade microtome (D.S.K microslicer DTK1000), which were allowed to recover for at least 30 min in oxygenated aCSF at 34 °C prior to recording, containing (in mM) 120 NaCl, 26 NaHCO$_3$, 1.25 NaH$_2$PO$_4$, 2.5 KCl, 5 HEPES, 14.6 glucose, 2 CaCl$_2$, 1 MgCl$_2$), pH 7.35–7.40, osmolarity 305–315 mOsm.

During recording, slices were perfused with aCSF at a flow rate of 2–3 ml/min. Targeted recordings were taken from tdTomato-expressing neurons using glass microelectrodes filled with a K-gluconate internal solution (containing 130 K-Gluconate, 5 NaCl, 1 EGTA, 10 HEPES, 5 Mg-ATP, 0.5 Na-GTP, 2 biocytin). Whole-cell recordings were acquired using a HEKA EPC10 amplifier with Patchmaster software at a sampling frequency of 20 kHz (HEKA Elektronik). A biophysical characterization of passive and active membrane properties was performed in current and voltage clamp modes, and the access resistance was monitored between recordings using a 10 mV voltage step protocol. Data were excluded if the access resistance changed >30% during recording.

The relative position of the recorded/labeled neurons in each slice was noted, and slices were fixed overnight in 4% PFA at 4 °C at the end of each experiment. Slices were immunoreacted with primary antibodies against tdTOM and TPH2, which were revealed the next day with secondary antibodies conjugated to Cy3 or Alexa 647. Biocytin was revealed with Avidin-Alexa A488 and the position of the filled neurons was assigned to the NRM or the LPGi, which could be determined based on the pattern of TPH2 immunoreactivity in the slice. Cells located outside of these two regions were not analyzed further.

## Behavioral assays

For specific DREADD-mediated activation of descending serotonergic pathways, an intersectional approach was used for labeling the lateral hindbrain neurons whereas direct labelling was used to label the medial neurons. Tph2-Cre mice received an intraspinal injection of AAV2retro.flex.FLPo. BFP, and 1 week later received bilateral injections into the ventral hindbrain with AAV1.FRT.hM3Dq. mCherry. Behavioral tests were performed after 10 days incubation time to allow the expression of the viral transgene. Before experiments mice were acclimatized to the behavioral setup for 1 hr in a room maintained at 20–22°C. For the Hargreaves plantar, cold plantar, electronic von Frey, and Rotarod assays, six measurements were taken for each time point and an average of these was reported. All measurements were taken from both hindlimbs of all animals. Alternatively, serotonergic neurons in the medial hindbrain were labelled by injection of AAV8.hsyn.flex.hM3Dq into the NRM (injection coordinates –6, 0, 5.9) and the same behavioral tests were performed with the same experimental design.

## Hargreaves

Sensitivity to heat stimuli was assessed with the Hargreaves plantar assay (IITC). Mice were placed on a pre-heated transparent platform set to a temperature of 30 °C, and the withdrawal latencies were recorded using a timed infrared heat source. A resting intensity of 5% and an active intensity of 20% was used for stimulations, and a maximum cutoff time of 32 s was set to avoid tissue damage. On average, naïve animals will withdraw to this stimulation at temperatures between 34 and 36°C (*Brenner et al., 2012*).

## Cold plantar assay

Mice were placed on a 5 mm borosilicate glass platform and were stimulated from beneath with dry ice pellets. The time taken to withdraw the paw was measured using a stopclock and a maximum stimulation time of 20 s was used to avoid tissue damage. The cold plantar assay generally produces a withdrawal when the temperature of the glass decreases 2 °C for naive animals (*Brenner et al., 2012*).

## von Frey

Von Frey thresholds were measured using an electronic von Frey algesiometer (IITC). Animals were adapted on a mesh surface, and the plantar surface of each paw was stimulated with a bendable plastic filament attached to a pressure sensitive probe. Pressure was applied to the plantar surface in a linear manner until the animal withdrew its paw, and the maximum pressure (the pressure at which the animal withdrew) was displayed on the device.

## Rotarod

Sensorimotor coordination was evaluated using an accelerating rotarod, and the time taken for animals to fall from the rotating barrel was recorded. The barrel rotated from 4 to 40 rpm over a period of 300 s, and increased speed constantly throughout each experiment. Values were discarded

if the animal jumped from the barrel, and if the animal jumped in >50% of trials for a given time point these data were discarded from the experiment. Two training sessions were given for all animals prior to the experiment being started to ensure a stable performance in the absence of treatment.

## Drug application

For DREADD activation experiments, clozapine-N-oxide (CNO, 2 mg/kg, Enzo life sciences, product number BML-NS105-0025) or vehicle controls were injected intraperitoneally, with the experimenter being blinded to the injected substance. Animals were tested directly before and 1–3 hr following i.p. injection. Stock CNO was dissolved in DMSO at 100 mg/ml and kept at room temperature, which was diluted 1:500 in sterile filtered saline immediately prior to injection with the volume of injected substance being 10 µl/g.

## Experimental design and statistical tests

All behavioral assays were performed twice on each animal so that they received an intraperitoneal injection of CNO on 1 day and vehicle injection on the other, with the experimenter being blinded to the treatment. Response latencies and thresholds before and following CNO and vehicle were compared using a repeated measures one-way ANOVA, a normal distribution was assumed for response latencies and thresholds and a Bonferroni post-hoc test was used to compare the mean withdrawal latencies and thresholds between CNO and vehicle-treated groups before and 1–3 hr after injection. To test whether any differences observed in the behavioral assays were sex-specific, groups were divided into male and female cohorts and the male and female post-vehicle and post-CNO latencies and thresholds were compared using multiple t-tests with Holm-Sidak correction. In control experiments in which CNO or vehicle were injected in the absence of hM3Dq, a two tailed paired t-test was performed to detect differences between withdrawal latencies and thresholds post-vehicle and post-CNO injection. Statistical significance was taken as $p < 0.05$.

To compare membrane properties between medial and lateral serotonergic neurons, a normal distribution in values was assumed and an unpaired two-tailed t-test was used to test for differences.

## Data collection, storage, and presentation

Data obtained from the quantification of images were collected and processed in Microsoft Excel and were presented using GraphPad Prism 5. All datapoints on graphs represent either a number or percentage of cells counted per animal, or as a response for each animal in behavioral assays. Representative images were produced in Affinity Photo and were annotated and arranged into figures in Affinity Designer. Raw data acquired in these experiments are uploaded to https://datadryad.org/stash and are available for download.

## Acknowledgements

We are grateful for funding provided by grants from the Swiss National Science Foundation (grant number 310030_197888) and by the clinical research priority programme (CRPP) "Pain – from phenotypes to mechanisms" of the Faculty of Medicine, University of Zurich, to HUZ, and a grant from the Olga Mayenfisch Stiftung to HW. We thank Louis Scheurer, Katharina Struckmeyer-Fichtel, and Isabelle Kellenberger for technical support.

## Additional information

### Funding

| Funder | Grant reference number | Author |
| --- | --- | --- |
| Schweizerischer Nationalfonds zur Förderung der Wissenschaftlichen Forschung | 310030_197888 | Hanns Ulrich Zeilhofer |
| Olga Mayenfisch Stiftung | | Hendrik Wildner |

| Funder | Grant reference number | Author |
| --- | --- | --- |

The funders had no role in study design, data collection and interpretation, or the decision to submit the work for publication.

## Author contributions

Robert Philip Ganley, Conceptualization, Formal analysis, Investigation, Methodology, Writing - original draft, Project administration, Writing - review and editing; Marilia Magalhaes de Sousa, Tugce Öztürk, Formal analysis, Investigation; Kira Werder, Formal analysis, Investigation, Writing - review and editing; Raquel Mendes, Matteo Ranucci, Investigation; Hendrik Wildner, Supervision, Funding acquisition, Project administration, Writing - review and editing; Hanns Ulrich Zeilhofer, Resources, Supervision, Funding acquisition, Writing - original draft, Writing - review and editing

## Author ORCIDs

Robert Philip Ganley http://orcid.org/0000-0001-8502-9870

## Ethics

Permission to perform these experiments was obtained from the Veterinäramt des Kantons Zürich (154/2018 and 063/2016).

## Decision letter and Author response

Decision letter https://doi.org/10.7554/eLife.78689.sa1
Author response https://doi.org/10.7554/eLife.78689.sa2

# Additional files

## Supplementary files

• MDAR checklist

## Data availability

All data generated or analysed during this study are included in the manuscript. Raw data acquired in these experiments are uploaded to https://datadryad.org/stash and are available for download.

The following dataset was generated:

| Author(s) | Year | Dataset title | Dataset URL | Database and Identifier |
| --- | --- | --- | --- | --- |
| Ganley RP | 2023 | Targeted Anatomical and Functional Identification of Antinociceptive and Pronociceptive Serotonergic Neurons that Project to the Spinal Dorsal Horn | https://dx.doi.org/10.5061/dryad.h70rxwdm9 | Dryad Digital Repository, 10.5061/dryad.h70rxwdm9 |

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
