## [Editor Report]

The study reveals anatomically and functionally distinct classes of serotonergic hindbrain neurons that are distinguished by their postsynaptic targets in the spinal cord as well as their contributions to behavioral responses to painful stimuli. These findings advance our understanding of the descending modulation of sensory processing in the spinal cord dorsal horn.

---

## [Decision Letter]

**Decision letter after peer review:**

Thank you for submitting your article "Targeted Anatomical and Functional Identification of Antinociceptive and Pronociceptive Serotonergic Neurons that Project to the Spinal Dorsal Horn" for consideration by *eLife*. Your article has been reviewed by 3 peer reviewers, and the evaluation has been overseen by a Reviewing Editor and Ronald Calabrese as the Senior Editor. The following individuals involved in the review of your submission have agreed to reveal their identity: Susan M Dymecki (Reviewer #2); Andrew Todd (Reviewer #3).

Essential revisions:

1. General concerns that should be addressed:

What the authors consider nucleus raphe magnus (NRM) should be clarified (e.g. Bregma levels). The presented tissue sections appear fairly caudal and will include multiple subtypes of serotonergic neurons, one of which may innervate the dorsal spinal cord and another that innervates the ventral spinal cord motor centers and likely influences motor gain. The more caudal, the greater enrichment for the latter subtype. It is likely that multiple functionally distinct subtypes of serotonergic neurons are being manipulated in the hM3Dq experiments targeting the presumed NRM. This should be considered, and published work in this area should be referenced. This is particularly relevant because if the presumed NRM injections are too caudal, most of the midline serotonergic cells would be those projecting to the ventral (not dorsal) horn and indeed this is what is seen in Figure 4.

The authors did not study sex differences although such differences are well-documented in serotonin signaling and pain perception. Sex differences should be addressed.

Additional control experiments that document the extent of the spread of the virus from the site of injection should be done.

It is not clear whether the projections are different between lamina I and lamina II, or whether these vary along the rostrocaudal axis. Since distinct spinal cord lamina is innervated differentially by specific subtypes of peripheral neurons – the authors should consider co-labeling descending neuronal projections with known laminar and DRG subtype markers or comment on this.

2. Detailed concerns as they relate to the figures and text that should be addressed:

Figure 1 and Table 5:

This figure and table, while interesting, are tangential to the narrative and overall claims of the paper and thus might better serve the manuscript as a supplemental figure rather than the lead figure.

As well, the conclusions drawn from this figure and table might be recast for stronger alignment with the data. It is concluded that "these data demonstrate that although TPH2-expressing hindbrain neurons can be directly infected with modified rabies viruses, they are largely underrepresented with monosynaptic tracing from the dorsal horn." While the latter is well supported by the data, the sentence construction suggests that the former direct labeling using SAD.RabiesΔG-eGFP (SAD-G) is effective, however, only a few cells were labeled (15 TPH2+ ventral hindbrain cells across 3 animals were marked using SAD.RabiesΔG-eGFP). Insertion of some caution might be helpful to the reader, perhaps stating that infectivity and/or labeling using SAD.RabiesΔG-eGFP occurs but is low, certainly as compared to CTb.

Given the differential efficacy of spinally injected AAV2retro.GFP to access LPGI versus NRM, it would be interesting to see if there is a bias for SAD.RabiesΔG-eGFP, yet no information is given as to the location of the few labeled cells summarized in panel C and Table 5. Further, panel B could be helped by including some anatomical landmarks or at least marking the ventral surface and midline.

– Panel C, y-axis labeling could be clearer, explicitly stating that plotted is the percentage of GFP+ cells that are also TPH2+.

– D: For the monosynaptic tracing experiment, it is stated that a similar total number of GFP+ cells are labeled in the hindbrain yet lacking are TPH2+ cells. This suggests that another population is captured that is not well represented in the direct labeling. Please clarify.

Table 5: columns 5 and 6 have the same headings. Presumably, column 6 is to reflect GFP neurons that fail to stain for TPH2. This needs correction.

Cre activity and presumed functional TVA levels in the desired dorsal horn cells should be demonstrated, as the conclusions around D-F depend on it. Further, the authors could consider performing monosynaptic tracing with AAV-Syn-TVA/G instead of R26-TVA, as this would have broader applicability to the work of many (few seem to use R26-TVA).

Figure 2:

D-E: An important comparison to the AAV2retro.eGFP data would be the quantification and presentation of CTb+ TPH2+ double-positive cells (like that in panel C but for CTb); as well as the co-injection of AAV2retro.eGFP with CTb, as co-injection can produce unpredicted confounding effects. Very few eGFP+ cells are discernible in the LPGi in the representative image. There are clear eGFP+ cells in the NRM. Can you please explain this discrepancy?

CTb could have some biases itself. Consider using another non-viral retrograde tracer.

AAV2retr.eGFP shows preferential infectivity of LPGi->spinal cord cells versus NRM->spinal cord cells; the claim of "highly susceptible" versus "resistant" to infection should be modified to better align with the data. 20% of labeled cells are in the NRM (thus not negligible, not really "resistant"), 80% in the LPGi.

A description of the areas denoted as "Other" is warranted, especially given their inclusion in the cell counts.

This figure could be supplemental.

Figure 3:

The main narrative of the manuscript seems to start with Figure 3. This figure should have an associated table with actual cell numbers.

This analysis should be repeated for the specific viruses used in Figures 6 and 7, where it is even more relevant to establish the differential infectivity and efficacy of cell-type-specific expression.

Some of the references in the Results text don't match up. Specifically on p. 8 "This resulted in TPH2-expressing neurons of the hindbrain being labelled with eGFP, particularly in the LPGi (Figure 2B)". This should presumably be referring to Figure 3B. And a bit later "…many neurons were labeled in the ipsilateral DRG (Figure 2C)". This should presumably be referring to Figure 3C.

Figure 3E. Why are only 77.5% TPH2+ when using a TPH2::Cre? This finding suggests that the Cre used is not as effective as one might like.

Figure 4:

G: For the extensive ventral spinal cord labeling, the authors should consider the midline serotonergic neurons likely responsible and what that might mean for the location of their injections (fairly caudal) and the multiple serotonergic neuron subtypes and functions likely perturbed in their DREADD experiments.

H-I: Quantification of tdTom-5-HT double positive boutons (puncta) would be helpful.

The use of markers in the spinal cord to identify the lamina would be helpful.

Figure 5:

This figure could be supplemental data. The present electrophysiological data is less informative considering the ultimate goal of the manuscript (identify functional subsets based on projection/transduction with AAVretro). The authors should consider performing ephys experiments w/ AAVretro injection in the spinal cord and patch from LPGi→spinal cord cells vs LPGi cells, and from NRM→spinal cord cells vs NRM cells. NRM cells labeled with AAVretro might be more similar to LPGi cells also labeled with AAVretro than other NRM cells. In essence, exploit the infectivity differences as classification and potentially reflective of functional differences rather than anatomy.

Please include the number of animals from which the recordings were derived.

The discussion here should include other biophysical properties that have been found to distinguish among caudal medullary serotonergic neuron subtypes (chemosensitivity, resting membrane potential, spontaneous spike frequency, response to thermal noxious stimuli, etc.).

Figures 6 and 7:

Characterization of virus efficacy, as in figure 3, needs to be performed here. Indeed, it is more important than that of Figure 3.

Needed is some demonstration of CNO-Dq triggered neuronal activation: FOS induction, or ephys measurements of the increase in excitability.

There is no control for CNO alone (in the absence of hM3Dq): Another cohort of mice needs to be injected with control AAVs and receive CNO. As it stands, the observed differences could be due to CNO off-target effects.

Of note, the von Frey withdrawal threshold of Figure 7 is much higher than in Figure 6, with CNO seemingly bringing the threshold level back to the level observed in Figure 6. Is this a calibration difference, observer difference, or other?

More detail is needed in the description of all three behavioral tests, e.g. temperature of effect.

Given that the midline serotonergic neuron subtype referred to as Tac1-Pet1 (Hennessy et al., 2017 J. Neurosci) projects to the ventral spinal cord as well as respiratory motor centers and likely is involved in shaping motor gain, its Dq manipulation could result in motor and respiratory phenotypes not revealed by the rotarod assay. For example, it could affect paw withdrawal speed which in turn might lower an apparent withdrawal latency, thus having little to do with mechanical sensitivity and nociception per se but rather motor effects.

P4 line 7: "lateral to the midline" – strictly, all cells would be covered by this definition, so I suggest altering the wording here.

P8 line 10: I think that they mean "retrograde transduction".

P9 last line: probably best to state that all coordinates are given in the order RC, ML, DV.

P11 line 13: "projected to both ipsilateral and contralateral dorsal horn" might be better.

P11: The last line cut off before reference [26]; Figure 5D. This panel is not mentioned in the results.

P12 para 2: Figure 5D is not mentioned in the text.

P15 para 1 of the Discussion: one difference between the anatomical results in this study and those of Gautier et al. (reference 26) is that Gautier et al. did not report any labelling in the ventral horn from injections into the NRM, whereas this was seen in the present study (Figure 4G). A brief discussion of this point would be helpful.

P18 para 2: This section of the discussion was confusing. It seems that a link is being made between the presence of 5-HT3 receptors and the occurrence of transsynaptic tracing with the rabies virus. However, this seems rather speculative, and it is suggested that this paragraph is either shortened or omitted.

P19 line 8: "the located serotonergic neurons" – presumably something is missing here.

P19 final para; the wording of the first sentence is rather awkward.

P20: last sentence: I suggest removing the sentence concerning intraspinal Fluorogold injections, as the data from these were not included in the study.

P22: "PBS with added salt". This is rather confusing since 8 g NaCl per litre is actually less than physiological saline (9 g per litre). Maybe this means that 8 g NaCl was added to this?

P24 para 3: "that were" in the first 2 lines of this para is in the wrong place. The experiments were not serotoninergic and the hindbrain wasn't labelled with eGFP.

Figure 2 legend: explain the arrows in E.

Figure 6 legend: spell out the differences indicated by asterisks (also in Figure 7)

Figure 7 legend: reference to the various parts is incorrect.

*Reviewer #1 (Recommendations for authors):*

1. Much of the data in the initial figures seems somewhat extraneous to the main points of the paper:

Figure 1A-C. Does rabies virus also label distinct hindbrain neuron populations? To test infection efficiency, why not also inject SAD-G directly into the hindbrain? Furthermore, this data could go in the supplement as rabies virus tracing strategies are not used in the paper.

Figure 2F-H. These data can go in the supplement. CTb tracing strategy is not utilized in this paper.

2. For the quantification, the authors indicate the number of animals examined, but I did not see the number of cells that were assessed. This would be useful to know. For example, Figure 3H. 78.3% of NRM cells were mCherry+ ; eGFP+. Out of how many cells total?

3. Figure 2G. Very few eGFP+ cells are discernible in the LPGi in the representative image. There are clear eGFP+ cells in the NRM. Can you explain this discrepancy?

4. Figure 3E. Why are only 77.5% TPH2+ when using a TPH2::Cre? This finding suggests that the Cre used is not as effective as one might like.

5. Figure 7D. NRM neurons project to lamina VII-IX, which are innervated by motor neurons. Authors should briefly discuss whether NRM activation had any effects on motor responses and whether this might contribute to the withdrawal response to von Frey filaments.

6. Page 11: The last line cut off before reference [26]; Figure 5D. This panel is not mentioned in the results.

*Reviewer #2 (Recommendations for authors):*

General recommendations that span multiple experiments/figures:

– Injection sites should be documented for each experiment, using a second virus or marker to document viral spread and actual targeting.

– Explanation is needed as to why all spinal cord injections seem to be at thoracic level 13 (T13), yet L2-L4 may make more sense given the hind limb behavioral phenotypes being assessed.

– What the authors consider nucleus raphe magnus (NRM) should be clarified (e.g. Bregma levels). The presented tissue sections appear fairly caudal and will include multiple subtypes of serotonergic neurons, one of which may innervate the dorsal spinal cord and another that innervates the ventral spinal cord motor centers and likely influences motor gain. The more caudal, the greater enrichment for the latter subtype. It is likely that multiple functionally distinct subtypes of serotonergic neurons are being manipulated in the hM3Dq experiments targeting the presumed NRM. This should be considered, as well as published work in this area should be referenced. It is particularly relevant because if the presumed NRM injections are too caudal, most of the midline serotonergic cells would be those projecting to the ventral (not dorsal) horn and indeed this is what they see in Figure 4.

– References that should be included as relates to medullary serotonergic neuron subtypes, functions, and hodology: Iceman, K.E. et al., 2013 J. Neurophysiol; Richerson, G.B. 2001 Respir. Physiol.; work of Gordon Mitchell; Brust et al., 2014 Cell Reports; Hennessy et al., 2017, J. Neuroscience; Depuy, S.D., et al., 2011 J. Neurosci; work of Peggy Mason.

– Suggest greater caution around the use of terms "antinociceptive" and "pronociceptive" (e.g. in the title). Are the behavioral tests performed at pain thresholds? More information for the reader is needed here.

– More details should be supplied around mouse lines: are the Cre driver lines BAC transgenics, IRES-cre knock-ins, or knock-in/knock-out lines? References are provided, but a direct provision of details would be helpful to the reader.

Specific recommendations related to each figure:

Figure 1 and Table 5:

– This figure and table, while interesting, are tangential to the narrative and overall claims of the paper and thus might better serve the manuscript as a supplemental figure rather than the lead figure.

– As well, the conclusions drawn from this figure and table might be recast for stronger alignment with the data. It is concluded that "these data demonstrate that although TPH2-expressing hindbrain neurons can be directly infected with modified rabies viruses, they are largely underrepresented with monosynaptic tracing from the dorsal horn." While the latter is well supported by the data, the sentence construction suggests that the former direct labeling using SAD.RabiesΔG-eGFP (SAD-G) is effective, however, only a few cells were labeled (15 TPH2+ ventral hindbrain cells across 3 animals were marked using SAD.RabiesΔG-eGFP). Insertion of some caution might be helpful to the reader, perhaps stating that infectivity and/or labeling using SAD.RabiesΔG-eGFP occurs but is low, certainly as compared to CTb.

– Given the differential efficacy of spinally injected AAV2retro.GFP to access LPGI versus NRM, it would be interesting to see if there is a bias for SAD.RabiesΔG-eGFP, yet no information is given as to the location of the few labeled cells summarized in panel C and Table 5. Further, panel B could be helped by including some anatomical landmarks or at least marking the ventral surface and midline.

– Panel C, y-axis labeling could be clearer, explicitly stating that plotted is the percentage of GFP+ cells that are also TPH2+.

– D: For the monosynaptic tracing experiment, it is stated that a similar total number of GFP+ cells are labeled in the hindbrain yet lacking are TPH2+ cells. This suggests that another population is captured that is not well represented in the direct labeling. Please clarify.

– Table 5: columns 5 and 6 have the same headings. Presumably, column 6 is to reflect GFP neurons that fail to stain for TPH2. This needs correction.

– Cre activity and presumed functional TVA levels in the desired dorsal horn cells should be demonstrated, as the conclusions around D-F depend on it. Further, the authors could consider performing monosynaptic tracing with AAV-Syn-TVA/G instead of R26-TVA, as this would have broader applicability to the work of many (few seem to use R26-TVA).

Figure 2:

– D-E: An important comparison to the AAV2retro.eGFP data would be the quantification and presentation of CTb+ TPH2+ double-positive cells (like that in panel C but for CTb); as well as the co-injection of AAV2retro.eGFP with CTb, as co-injection can produce unpredicted confounding effects.

– CTb could have some biases itself. Consider using another non-viral retrograde tracer.

– AAV2retr.eGFP shows preferential infectivity of LPGispinal cord cells versus NRMspinal cord cells; the claim of "highly susceptible" versus "resistant" to infection should be modified to better align with the data. 20% of labeled cells are in the NRM (thus not negligible, not really "resistant"), 80% in the LPGi.

– A description of the areas denoted as "Other" is warranted, especially given their inclusion in the cell counts.

– This figure could be supplemental.

Figure 3:

– The main narrative of the manuscript seems to start with Figure 3.

– Should have an associated table of actual cell numbers.

– This analysis should be repeated for the specific viruses used in Figures 6 and 7, where it is even more relevant to establish the differential infectivity and efficacy of cell-type-specific expression.

– Some of the references in the Results text don't match up. Specifically on p. 8 "This resulted in TPH2-expressing neurons of the hindbrain being labelled with eGFP, particularly in the LPGi (Figure 2B)". This should presumably be referring to Figure 3B. And a bit later "…many neurons were labeled in the ipsilateral DRG (Figure 2C)". This should presumably be referring to Figure 3C.

Figure 4:

– G: For the extensive ventral spinal cord labeling, the authors should consider the midline serotonergic neurons likely responsible and what that might mean for the location of their injections (fairly caudal) and the multiple serotonergic neuron subtypes and functions likely perturbed in their DREADD experiments.

– H-I: Quantification of tdTom-5-HT double positive boutons (puncta) would be helpful.

– Use of markers in the spinal cord to identify the lamina would be helpful.

Figure 5:

– Could be supplemental data. The present electrophysiological data is less informative

considering the ultimate goal of the manuscript (identify functional subsets based on projection/transduction with AAVretro). Consider performing ephys experiments w/ AAVretro injection in the spinal cord and patch from LPGi→spinal cord cells vs LPGi cells, and from NRM→spinal cord cells vs NRM cells. NRM cells labeled with AAVretro might be more similar to LPGi cells also labeled with AAVretro than other NRM cells. In essence, exploit the infectivity differences as classification and potentially reflective of functional differences rather than anatomy.

– Include the number of animals from which the recordings were derived.

– Discussion here should include other biophysical properties that have been found to distinguish among caudal medullary serotonergic neuron subtypes (chemosensitivity, resting membrane potential, spontaneous spike frequency, response to thermal noxious stimuli, etc.)

Figures 6 and 7:

– Characterization of virus efficacy, as in figure 3, needs to be performed here. Indeed, it is more important than that of Figure 3.

– Needed is some demonstration of CNO-Dq triggered neuronal activation: FOS induction, or ephys measurements of the increase in excitability.

– There is no control for CNO alone (in the absence of hM3Dq): Another cohort of mice needs to be injected with control AAVs and receive CNO. As it stands, the observed differences could be due to CNO off-target effects.

– Of note, the von Frey withdrawal threshold of Figure 7 is much higher than in Figure 6, with CNO seemingly bringing the threshold level back to the level observed in Figure 6. Is this a calibration difference, observer difference, or other?

– More detail is needed in the description of all three behavioral tests, e.g. temperature of effect.

– Given that the midline serotonergic neuron subtype referred to as Tac1-Pet1 (Hennessy et al., 2017 J. Neurosci) projects to the ventral spinal cord as well as respiratory motor centers and likely is involved in shaping motor gain, its Dq manipulation could result in motor and respiratory phenotypes not revealed by the rotarod assay. For example, it could affect paw withdrawal speed which in turn might lower an apparent withdrawal latency, thus having little to do with mechanical sensitivity and nociception per se but rather motor effects.

*Reviewer #3 (Recommendations for authors):*

I have a few specific suggestions for improving the article:

P4 line 7: "lateral to the midline" – strictly, all cells would be covered by this definition, so I suggest altering the wording here.

P8 line 10: I think that they mean "retrograde transduction".

P9 last line: probably best to state that all coordinates are given in the order RC, ML, DV.

P11 line 13: "projected to both ipsilateral and contralateral dorsal horn" might be better.

P12 para 2: Figure 5D is not mentioned in the text.

P15 para 1 of the Discussion: one difference between the anatomical results in this study and those of Gautier et al. (reference 26) is that Gautier et al. did not report any labelling in the ventral horn from injections into the NRM, whereas this was seen in the present study (Figure 4G). A brief discussion of this point would be helpful.

P18 para 2: I found this section of the discussion rather confusing. It seems that a link is being made between the presence of 5-HT3 receptors and the occurrence of transsynaptic tracing with the rabies virus. However, this seems rather speculative, and I suggest that this paragraph is either shortened or omitted.

P19 line 8: "the located serotonergic neurons" – presumably something is missing here.

P19 final para; the wording of the first sentence is rather awkward.

P20: last sentence: I suggest removing the sentence concerning intraspinal Fluorogold injections, as the data from these were not included in the study.

P22: "PBS with added salt". This is rather confusing since 8 g NaCl per litre is actually less than physiological saline (9 g per litre). Maybe this means that 8 g NaCl was added to this?

P24 para 3: "that were" in the first 2 lines of this para is in the wrong place. The experiments were not serotoninergic and the hindbrain wasn't labelled with eGFP.

Figure 2 legend: explain the arrows in E.

Figure 6 legend: spell out the differences indicated by asterisks (also in Figure 7)

Figure 7 legend: reference to the various parts is incorrect.

---

## [Author Response]

Essential revisions:1. General concerns that should be addressed:What the authors consider nucleus raphe magnus (NRM) should be clarified (e.g. Bregma levels). The presented tissue sections appear fairly caudal and will include multiple subtypes of serotonergic neurons, one of which may innervate the dorsal spinal cord and another that innervates the ventral spinal cord motor centers and likely influences motor gain. The more caudal, the greater enrichment for the latter subtype. It is likely that multiple functionally distinct subtypes of serotonergic neurons are being manipulated in the hM3Dq experiments targeting the presumed NRM. This should be considered, and published work in this area should be referenced. This is particularly relevant because if the presumed NRM injections are too caudal, most of the midline serotonergic cells would be those projecting to the ventral (not dorsal) horn and indeed this is what is seen in Figure 4.

The midline injections certainly include the serotonergic regions beyond the NRM, and some of these are seen to project to the ventral horn (Figure 4G, now figure 3G). We have included this as an alternative explanation for the data presented in figure 7, which is also now a part of the discussion (page 19 line 11). We have also altered the wording in the Results section to make it clear that these injections include the NRM, but may extend to medial serotonergic neurons rostral and caudal to the injection site (page 11 line 9, and page 16 line 1). Despite these limitations, the selectivity of the effect on mechanical sensitivity indicates that this is likely a sensory effect rather than solely an effect on motor reflexes.

The authors did not study sex differences although such differences are well-documented in serotonin signaling and pain perception. Sex differences should be addressed.

This is an important point, and we thank the reviewers for drawing our attention to this. For the behavioral experiments we have included figure supplements to see whether the behavioral effects seen are also present in both males and females (figure 5 figure supplement 3, and figure 6 figure supplement 2). These results are discussed in the Results section (page 14 line 23, and page 16 line 16). We have also added more animals to the behavioral experiments to balance the number of female and male animals. We find that the behavioral phenotypes observed in the chemogenetic experiments are present in both male and female mice.

Additional control experiments that document the extent of the spread of the virus from the site of injection should be done.

For all experiments in which the midline serotonergic neurons were labelled by direct injection, we confirm that there is no spread to the lateral serotonergic neurons. In these experiments we did observe spread to other medial serotonergic hindbrain nuclei, such as the raphe pallidus and raphe obscurus, we have included all injection sites used in the chemogenetic activation experiments as a figure supplement (figure 6 figure supplement 1). We accept that there are other populations captured in the injections that likely influence the behavior, and have included this limitation as part of the discussion (page 19 line 11).

It is not clear whether the projections are different between lamina I and lamina II, or whether these vary along the rostrocaudal axis. Since distinct spinal cord lamina is innervated differentially by specific subtypes of peripheral neurons – the authors should consider co-labeling descending neuronal projections with known laminar and DRG subtype markers or comment on this.

We have performed immunostaining experiments to assess which laminae are innervated by the axon terminals. We used PKCg staining to mark lamina IIi and CGRP staining to delineate laminae I-IIo, both of which are commonly used in spinal cord anatomical studies. We were able to determine the precise laminae innervated by the medial versus the lateral serotonergic neurons. Lateral serotonergic neurons of the LPGi are the only serotonergic neurons that densely innervate the superficial laminae (I-IIo), whereas the medial serotonergic neurons project to ventral laminae (ventral to the lamina IIi-III border). Text discussing these results has been added to the Results section (page 11 line 22, and page 12 line 9).

2. Detailed concerns as they relate to the figures and text that should be addressed:Figure 1 and Table 5:This figure and table, while interesting, are tangential to the narrative and overall claims of the paper and thus might better serve the manuscript as a supplemental figure rather than the lead figure.

We agree that this does not fit together with the rest of the manuscript and would be more suitable as a supplemental figure (now figure 1 figure supplement 4). Since we are discussing retrograde tracing strategies in this section it seems appropriate to include it here. The text has also been moved to a later part of the results (page 6 line 21).

As well, the conclusions drawn from this figure and table might be recast for stronger alignment with the data. It is concluded that "these data demonstrate that although TPH2-expressing hindbrain neurons can be directly infected with modified rabies viruses, they are largely underrepresented with monosynaptic tracing from the dorsal horn." While the latter is well supported by the data, the sentence construction suggests that the former direct labeling using SAD.RabiesΔG-eGFP (SAD-G) is effective, however, only a few cells were labeled (15 TPH2+ ventral hindbrain cells across 3 animals were marked using SAD.RabiesΔG-eGFP). Insertion of some caution might be helpful to the reader, perhaps stating that infectivity and/or labeling using SAD.RabiesΔG-eGFP occurs but is low, certainly as compared to CTb.

We appreciate that the total number of cells labelled in the hindbrain in the rabies tracing experiments is relatively low when compared to the other tracing experiments in the manuscript and have included the total cell counts in table 2. In our hands, the rabies virus tracing labels far fewer cells in supraspinal regions when compared to AAV2retro or Ctb, possibly because rabies viruses are more sensitive to inactivation than AAVs or CTb. Additionally, the viral titre typically achieved with rabies virus production is far lower than those that can be obtained with AAV production (1E9 vg/ml vs 1E13 vg/ml respectively) We have added text to the discussion to highlight that this is a small sample, the interpretation of which should be approached with caution.

“However, both direct and transsynaptic rabies labelling were far less sensitive than the CTb or AAV2retro tracing, making interpretation of these findings complicated due to the comparatively small sample sizes” (page 21 line 9).

However, we conclude that the overall interpretation that serotonergic neurons are not reliably labelled with transsynaptic tracing is correct, since almost half of the labelled RVM neurons contained TPH2 in the direct labelling experiments whereas virtually none were labelled with the transsynaptic tracing.

Given the differential efficacy of spinally injected AAV2retro.GFP to access LPGI versus NRM, it would be interesting to see if there is a bias for SAD.RabiesΔG-eGFP, yet no information is given as to the location of the few labeled cells summarized in panel C and Table 5. Further, panel B could be helped by including some anatomical landmarks or at least marking the ventral surface and midline.

The figure (now figure 1 figure supplement 1) has been modified to include the ventral surface of the hindbrain, the midline, and the different regions of the hindbrain are labelled with text. We did not notice any particular enrichment/absence of GFP-labelled cells from any of these areas, although with the small sample size this is difficult to determine.

– Panel C, y-axis labeling could be clearer, explicitly stating that plotted is the percentage of GFP+ cells that are also TPH2+.

This axis has been changed to %GFP+ cells expressing TPH2.

– D: For the monosynaptic tracing experiment, it is stated that a similar total number of GFP+ cells are labeled in the hindbrain yet lacking are TPH2+ cells. This suggests that another population is captured that is not well represented in the direct labeling. Please clarify.

Thank you for bringing this to our attention. There is presumably an increased efficiency in labelling other non-serotonergic neurons in the transsynaptic tracing, although besides the fact that they do not express TPH2 we cannot say much more about the identity of these neurons. Possibly this increased tracing is from descending neurons that innervate spinal neurons with a high synaptic density. We have added text to our discussion to highlight this.

“Since similar numbers of hindbrain neurons were labelled using both approaches in addition to the deficiency in labelling serotonergic neurons with monosynaptic tracing, this could indicate an increased labelling efficiency of other neurons. Potentially this could be related to the synaptic density from these neurons to the starter population” (page 21 line 5)

Table 5: columns 5 and 6 have the same headings. Presumably, column 6 is to reflect GFP neurons that fail to stain for TPH2. This needs correction.

The heading for column 6 should read GFP neurons not expressing TPH2. This has been corrected (now table 2).

Cre activity and presumed functional TVA levels in the desired dorsal horn cells should be demonstrated, as the conclusions around D-F depend on it. Further, the authors could consider performing monosynaptic tracing with AAV-Syn-TVA/G instead of R26-TVA, as this would have broader applicability to the work of many (few seem to use R26-TVA).

The Cre expression in the HoxB8^cre^ mouse is transient in all spinal neurons caudal to spinal segment C7, but the expression of TVA is stable after the recombination of the loxP sites in the R26-TVA allele (similar to Witschi et al., 2010, Figures 2 and 3). We agree that the injection of AAV-Syn-TVA/G would be an interesting experiment more similar to those performed by others. However, in these experiments we wanted a way to visualise the starter cells with a fluorophore (mCherry), to be certain that all GFP+ neurons in the hindbrain were transsynaptically traced from the spinal cord. This meant that we would have had to inject two different helper viruses (owing to the genome capacity of AAVs, it is not possible for TVA, rabies G and a fluorophore to be contained in a single vector) and in our experience, injecting two AAVs can lead to greatly reduced efficiency of one of the viruses. Since we needed all components to be expressed in the same neurons for the tracing, we chose to use the transgenic mice to provide one of these components, so the other two could be provided by a single type of AAV.

Figure 2:D-E: An important comparison to the AAV2retro.eGFP data would be the quantification and presentation of CTb+ TPH2+ double-positive cells (like that in panel C but for CTb); as well as the co-injection of AAV2retro.eGFP with CTb, as co-injection can produce unpredicted confounding effects. Very few eGFP+ cells are discernible in the LPGi in the representative image. There are clear eGFP+ cells in the NRM. Can you please explain this discrepancy?

Thank you for this suggestion, this will allow a more informative comparison between CTb and AAV2retro.eGFP retrograde labelling. We have included a figure supplement to show the proportion of TPH2 -labelled neurons with the CTb labelling (figure 1 figure supplement 2). As part of the main figure (now figure 1F) we have included a graph directly comparing the CTb and eGFP labelling of TPH2+ hindbrain neurons.

We accept that issues with co-injecting tracers are important concerns. In our experience these confounding effects are seen with the labelling with AAVs when more than one type of vector is injected, but this eGFP labelling appears similar between Figure 1C and in the new graph Figure 1F. This suggests that the inclusion of CTb hasn’t affected the labelling with AAV2retro.

The graphs that illustrate the proportions of neurons labelled with CTb and eGFP for all neurons and the different TPH2+ groups have been moved to a figure supplement (figure 1 supplement 3).

The green cells labelled in figure 2E (now figure 1 —figure supplement 2B) are the CTb-labelled neurons, which are found in both the NRM and the LPGi. The eGFP neurons in Figure 2B and G (now figure 1B and E) are also found throughout the RVM including the NRM, however those that contain TPH2 are mainly found in the LPGi.

CTb could have some biases itself. Consider using another non-viral retrograde tracer.

The CTb labelling likely has some biases in retrograde labelling and we agree that it would be interesting to compare all of the commonly used retrograde tracers. However, we feel this goes beyond the scope of this study, as the main finding from these experiments was the selectivity of the AAV2retro serotype, which was revealed when compared to the CTb labelling. The rest of the study utilizes this AAV2retro selectivity rather than further investigating the differences in labelling efficiency between various retrograde tracers.

AAV2retr.eGFP shows preferential infectivity of LPGi->spinal cord cells versus NRM->spinal cord cells; the claim of "highly susceptible" versus "resistant" to infection should be modified to better align with the data. 20% of labeled cells are in the NRM (thus not negligible, not really "resistant"), 80% in the LPGi.

We agree that the NRM neurons are not completely resistant to transduction with the AAV2r, and have changed all parts of the text describing these neurons as highly susceptible and resistant to “less sensitive” or “largely resistant”.

A description of the areas denoted as "Other" is warranted, especially given their inclusion in the cell counts.

Areas denoted as “Other” refer to all areas outside of the NRM and LPGi, which were readily identifiable with dense TPH2 immunostaining. We have provided a supplement to the figure 1 (figure 1 supplement 1) denoting the areas we classified as NRM, LPGi, and other. We also highlight some examples of retrogradely-labelled serotonergic neurons serotonergic that would be classified as other with markers.

This figure could be supplemental.

We have considered making this a supplemental figure, but we believe that the selectivity of the AAV2retro for the lateral neurons is a key part of the manuscript, and provides the basis for much of what is shown in later figures.

Figure 3:The main narrative of the manuscript seems to start with Figure 3. This figure should have an associated table with actual cell numbers.

We have provided 2 additional tables to accompany the results in figure 3 D E and H (now figure 2 D, E, and H) with the cell counts that were used to produce the graphs These are Tables 3 and 4 in the resubmitted manuscript.

This analysis should be repeated for the specific viruses used in Figures 6 and 7, where it is even more relevant to establish the differential infectivity and efficacy of cell-type-specific expression.

Most of the elements of the vectors are identical (such as serotype, promoter and recombinase dependency) besides the expressed transgene, and therefore it is expected that the labelling specificity should be the same. We have performed quantifications of the labelled cells on the tissues from animals used in additional behavioral experiments, which demonstrate that this labelling is similar. These graphs are included as figure supplements (figure 5 figure supplement 5 and figure 6 figure supplement 3) and are discussed in the text of the Results section (page 15 line 8, and page 16 line 23).

Some of the references in the Results text don't match up. Specifically on p. 8 "This resulted in TPH2-expressing neurons of the hindbrain being labelled with eGFP, particularly in the LPGi (Figure 2B)". This should presumably be referring to Figure 3B. And a bit later "…many neurons were labeled in the ipsilateral DRG (Figure 2C)". This should presumably be referring to Figure 3C.

Thank you for noticing these oversights. These have been changed in the text to accurately refer to the appropriate figure.

Figure 3E. Why are only 77.5% TPH2+ when using a TPH2::Cre? This finding suggests that the Cre used is not as effective as one might like.

This is likely due to the levels of TPH2 in these cells being too low to be detected with our immunostaining protocols, or the expression of cre is present in some non-serotonergic neurons within the hindbrain. However, when we look at the spinal cords, most of the labelled boutons are positive for serotonin, so we are confident that the majority of the findings from our behavioral and anatomical experiments are due to activation/labelling of serotonergic neurons, rather than non-serotonergic neurons.

Figure 4:G: For the extensive ventral spinal cord labeling, the authors should consider the midline serotonergic neurons likely responsible and what that might mean for the location of their injections (fairly caudal) and the multiple serotonergic neuron subtypes and functions likely perturbed in their DREADD experiments.

The labelling experiments do indeed show axon labelling in the ventral horn, and we agree that this is probably due to neurons projecting to the ventral spinal cord, or the spread of the virus to other caudal serotonergic nuclei around the midline. While it was not possible to completely limit the spread of the virus to the NRM with our approach, it was possible to only label neurons in the medial hindbrain without capturing those in the lateral hindbrain. This is mentioned in the Results section for the labelling and activation experiments (page 11 line 9, and page 16 line 1).

The behavioral data for figure 7 could also be interpreted as an alteration in the sensitivity of motor circuits in the ventral horn that mediate reflexes and movement. We have included this interpretation as part of the discussion in the revised manuscript

“The activation of the midline serotonergic neurons influences multiple functionally different groups within the injection site, including the NRM, ROb, and RPa. These more caudal nuclei are known to affect the motor system and respiratory functions. Within the sensory assays tested, there was a rather selective change in mechanical responses and unaltered thermal responses, indicating that the effect was modality selective and unlikely to be due to motor effects alone. However, it cannot be excluded that serotonergic neurons in these distinct midline regions contribute to the reduced mechanical thresholds. Further studies of sensory function utilizing intersectional strategies to precisely capture and manipulate the midline populations will help to validate the present findings.” (page 19 line 11)

However, as this altered behavior was only observed in the von Frey assay, rather than all assays, we are convinced that this is not solely caused by an effect on the motor system and is likely due to a sensory phenotype.

H-I: Quantification of tdTom-5-HT double positive boutons (puncta) would be helpful.

The formal quantification of 5-HT in the boutons would be interesting, however this is challenging due to the difficulty in objectively distinguishing the axon boutons from the rest of the axon. This is essential as the boutons would be expected to contain 5-HT whereas the intervaricose portions would not. Without a specific marker for these, it can be difficult to give an unbiased count required for the quantification. However, we feel the quantification of TPH2 at the cell soma (which can be objectively defined by NeuN staining), demonstrates that the majority of the cells labelled are serotonergic (TPH2+), and the images in figure 4 (now figure 3) show that there is an overlap in 5-HT and tdTOM staining.

The use of markers in the spinal cord to identify the lamina would be helpful.

We have included additional images from tissues containing labelled axons from the NRM and LPGi labelling experiments, which are immunostained for PKCg and CGRP to label laminae IIi and I-IIo respectively (figure 3 figure supplement 1). These images highlight that the projections from the LPGi terminate mostly in laminae I-IIo, whereas the majority of the axons from the midline neurons terminate ventral to the IIi/III border (defined by the PKCg plexus).

Figure 5:This figure could be supplemental data. The present electrophysiological data is less informative considering the ultimate goal of the manuscript (identify functional subsets based on projection/transduction with AAVretro). The authors should consider performing ephys experiments w/ AAVretro injection in the spinal cord and patch from LPGi→spinal cord cells vs LPGi cells, and from NRM→spinal cord cells vs NRM cells. NRM cells labeled with AAVretro might be more similar to LPGi cells also labeled with AAVretro than other NRM cells. In essence, exploit the infectivity differences as classification and potentially reflective of functional differences rather than anatomy.

These suggested electrophysiology experiments are intriguing and would be of great interest to many researchers in this field. We would be interested in following these up in a separate study. However, we feel that these experiments would be beyond the scope of this manuscript. The main purpose of this study primarily was to compare the anatomical properties and functional roles of lateral vs medial serotonergic hindbrain neurons, and the use of the AAV2retro was the most convenient tool to enable this. As we observed, many of the intrinsic electrophysiological properties of neurons in these areas are very similar, and are less likely to be responsible for the apparent behavioral differences seen between figures 6 and 7 (now figures 5 and 6). We assume that this is more likely a result of the anatomical features, such as soma location and axon termination pattern, which we document in more detail. Additionally, we attempted the suggested retrograde labelling of LPGi neurons for targeted recordings in preliminary experiments, and we generally found very few suitable cells for recording using this approach. We used the anatomical location of the cells in order to get a suitable number of recordings from fewer animals.

Please include the number of animals from which the recordings were derived.

LPGi recordings from 3 animals

NRM recordings from 4 animals

This information has been added to the text in the figure legend for figure 4 and to table 5.

The discussion here should include other biophysical properties that have been found to distinguish among caudal medullary serotonergic neuron subtypes (chemosensitivity, resting membrane potential, spontaneous spike frequency, response to thermal noxious stimuli, etc.).

We thank the reviewers for the suggestions, these are features that should be included in the discussion. We have included a short paragraph to describe these features and compare them with our results and mention the different approaches (in vivo vs in vitro) as a likely difference between our findings and those of others (page 20 line 8).

Figures 6 and 7:Characterization of virus efficacy, as in figure 3, needs to be performed here. Indeed, it is more important than that of Figure 3.

We have performed quantifications of these injections similarly to the analyses performed in figure 3 and 4. We obtain very similar results to those obtained in the hM3Dq experiments and these are included in a figure supplement to both figure 6 and 7 (now figures 5 and 6). These are figure 5 figure supplement 5 and figure 6 figure supplement 3.

Needed is some demonstration of CNO-Dq triggered neuronal activation: FOS induction, or ephys measurements of the increase in excitability.

We have used the tissues from the additional experiments to assess the proportion of hM3Dq-expressing neurons activated by i.p. CNO injection that upregulate c-fos. This is included as a figure supplement to figure 6 (now figure 5 figure 5 figure supplement 1).

There is no control for CNO alone (in the absence of hM3Dq): Another cohort of mice needs to be injected with control AAVs and receive CNO. As it stands, the observed differences could be due to CNO off-target effects.

We have performed experiments to assess the effects of CNO on the assays that were reported (Hargreaves, cold plantar, von Frey and Rotarod) and find that the CNO alone does not appear to have an influence. These are included as a supplement to figure 6 (now figure 5) (figure 5 —figure supplement 4) and are discussed in the text (page 15 line 3).

Of note, the von Frey withdrawal threshold of Figure 7 is much higher than in Figure 6, with CNO seemingly bringing the threshold level back to the level observed in Figure 6. Is this a calibration difference, observer difference, or other?

This could be due to the animals being different ages between these experiments, with the animals that received two injections (figure 6, now figure 5) being older than the animals that only received one (figure 7, now figure 6). This could in part explain some of the differences seen in the von Frey thresholds.

More detail is needed in the description of all three behavioral tests, e.g. temperature of effect.

Additional information has been added to the Methods section, and reference is made to Brenner et al. 2012, which is the original paper where the temperature tests are described in detail, and which we base most of our tests on (page 29 line 20 and line 25).

Given that the midline serotonergic neuron subtype referred to as Tac1-Pet1 (Hennessy et al., 2017 J. Neurosci) projects to the ventral spinal cord as well as respiratory motor centers and likely is involved in shaping motor gain, its Dq manipulation could result in motor and respiratory phenotypes not revealed by the rotarod assay. For example, it could affect paw withdrawal speed which in turn might lower an apparent withdrawal latency, thus having little to do with mechanical sensitivity and nociception per se but rather motor effects.

We agree that these phenotypes are likely to be influenced in the experiments of figure 7, similar to those observed in Hennessy et al. 2017. We have included this as an alternative explanation to our findings in the Discussion section of the paper (page 19 line 11). Despite these technical limitations related to specificity, we are convinced that our observed reduction in mechanical sensitivity are, in part, the result of a sensory phenotype since other withdrawal behaviors were unchanged in the same experiments.

P4 line 7: "lateral to the midline" – strictly, all cells would be covered by this definition, so I suggest altering the wording here.

The wording here has been changed to within the lateral paragigantocellularis, to make it clear that this is where the lateral labelled cells are located (page 5 line 12).

P8 line 10: I think that they mean "retrograde transduction".

Transsynaptic transduction has been changed to retrograde transduction (page 6 line 20).

P9 last line: probably best to state that all coordinates are given in the order RC, ML, DV.

This detail has been added to the methods section where stereotaxic brain injections are described (page 24 line 8).

P11 line 13: "projected to both ipsilateral and contralateral dorsal horn" might be better.

This has been changed (page 12 line 1).

P11: The last line cut off before reference [26]; Figure 5D. This panel is not mentioned in the results.

This has been corrected.

P12 para 2: Figure 5D is not mentioned in the text.

This is now referred to in the text.

P15 para 1 of the Discussion: one difference between the anatomical results in this study and those of Gautier et al. (reference 26) is that Gautier et al. did not report any labelling in the ventral horn from injections into the NRM, whereas this was seen in the present study (Figure 4G). A brief discussion of this point would be helpful.

This is a notable difference from our results, and we thank the reviewers for this observation. We have included this in the Discussion section of the paper (page 17 line 15), explaining that the experiments of Gautier et al. were performed in the rat, and presumably the larger brain size makes it possible to target the NRM with greater specificity. Alternatively, since this paper focused solely on the dorsal horn, there observations from the ventral horn may not have been reported.

P18 para 2: This section of the discussion was confusing. It seems that a link is being made between the presence of 5-HT3 receptors and the occurrence of transsynaptic tracing with the rabies virus. However, this seems rather speculative, and it is suggested that this paragraph is either shortened or omitted.

This section is rather speculative, as the assumption is that monosynaptic tracing does not capture serotonergic neurons because they do not commonly form many fast synaptic contacts in the dorsal horn. However, the 5-HT3 receptor may not be the sole receptor expressed in fast synapses, but this was assumed since it is the only receptor that would allow ionotropic signaling. This section has been removed. (page 21 line 19)

P19 line 8: "the located serotonergic neurons" – presumably something is missing here.

This has been changed to “the serotonergic neurons of the lateral hindbrain” (page 22 line 7).

P19 final para; the wording of the first sentence is rather awkward.

“Primarily, this data establishes descending serotonergic neurons originating in the LPGi as antinociceptive during acute activation” has been changed to “Primarily, our data demonstrate that serotonergic neurons of the lateral hindbrain are antinociceptive when activated”(page 22 line 16).

P20: last sentence: I suggest removing the sentence concerning intraspinal Fluorogold injections, as the data from these were not included in the study.

All mention of Fluorogold has been removed from the manuscript as no data is shown from these experiments.

P22: "PBS with added salt". This is rather confusing since 8 g NaCl per litre is actually less than physiological saline (9 g per litre). Maybe this means that 8 g NaCl was added to this?

«PBS added salt» is now changed to PBS + 8g/L NaCl throughout the text.

P24 para 3: "that were" in the first 2 lines of this para is in the wrong place. The experiments were not serotoninergic and the hindbrain wasn't labelled with eGFP.

This sentence has been rewritten and now reads “To determine whether serotonergic neurons were labelled using modified rabies virus tracing, eGFP labelled neurons in the hindbrain were identified and immunostained with TPH2” (page 27 line 15).

Figure 2 legend: explain the arrows in E.

The arrows in Figure 2 E (now figure 1 figure supplement 2B) highlight the many TPH2+ neurons in the NRM that are labelled with CTb. This has been added to the figure legend.

Figure 6 legend: spell out the differences indicated by asterisks (also in Figure 7)

The comparisons that are significantly different are now described in detail in both figure 6 and 7 (now figures 5 and 6).

Figure 7 legend: reference to the various parts is incorrect.

Apologies for this oversight, parts of this figure were rearranged, and the references were not updated. This has now been corrected.

Reviewer #1 (Recommendations for authors):1. Much of the data in the initial figures seems somewhat extraneous to the main points of the paper:Figure 1A-C. Does rabies virus also label distinct hindbrain neuron populations? To test infection efficiency, why not also inject SAD-G directly into the hindbrain? Furthermore, this data could go in the supplement as rabies virus tracing strategies are not used in the paper.Figure 2F-H. These data can go in the supplement. CTb tracing strategy is not utilized in this paper.

This would be an interesting experiment to address whether certain hindbrain neurons are resistant to infection with rabies viruses. Since the rabies tracing is not the main focus of the manuscript, and the experiments that were done assess how effectively this tracing approach captures serotonergic neurons via their axon terminals in the spinal cord, we feel that this would be beyond the scope of the study. However, we have added text in the discussion to explore this possibility (page 20 line 22).

The Figure 1 has now been moved to a supplemental figure (Figure 1 figure supplement 4).

Although the CTb tracing was not utilized in the rest of the manuscript, we believe that these data are important for the main narrative of the study, as we use the selectivity of the AAV2retro to influence particular serotonergic neurons. The comparison with the CTb highlights this selectivity and therefore we feel it belongs in one of the main figures. The CTb tracing alone has been moved to a supplementary figure (figure 1 figure supplement 2).

2. For the quantification, the authors indicate the number of animals examined, but I did not see the number of cells that were assessed. This would be useful to know. For example, Figure 3H. 78.3% of NRM cells were mCherry+ ; eGFP+. Out of how many cells total?

Thank you for the suggestion. We have included these data in tables (table 3 and table 4).

3. Figure 2G. Very few eGFP+ cells are discernible in the LPGi in the representative image. There are clear eGFP+ cells in the NRM. Can you explain this discrepancy?

In this figure there are eGFP neurons labelled in the LPGi, but there are also many eGFP neurons in the NRM. However, most of these do not express TPH2 (unlike the CTB labelled neurons, which frequently co-express TPH2). Many of the differences we see are when considering the TPH2+ neurons alone (as in Figure 1F, and figure 1, figure supplement 3), which may not be so apparent in Figure 2H (now figure 1E) where many TPH2-ve cells are also labelled.

4. Figure 3E. Why are only 77.5% TPH2+ when using a TPH2::Cre? This finding suggests that the Cre used is not as effective as one might like.

This is either due to the labelled neurons containing a low level of TPH2 that is undetectable with our immunostaining protocol, or alternatively due to expression of TPH2::cre in non-serotonergic neurons. However, as the majority of these neurons contain detectable TPH2, have broad action potentials and slow afterhyperpolarizations typical for serotonergic neurons, and the majority of labelled axon terminals in the spinal cord contain serotonin, this indicates that the behavioral effects seen later in this study are likely due to the activation of serotonergic projection neurons rather than off target effects.

5. Figure 7D. NRM neurons project to lamina VII-IX, which are innervated by motor neurons. Authors should briefly discuss whether NRM activation had any effects on motor responses and whether this might contribute to the withdrawal response to von Frey filaments.

Thank you for drawing our attention to these alternative interpretations of the data. We have added this alternative explanation to the discussion (page 19 line 11). We are convinced that in these experiments the reduced thresholds to von Frey stimulation are at least in part due to a sensory phenotype as it specifically affects mechanical sensitivity, rather than a global effect on all withdrawal responses, which would be expected if the effects were solely the result of a motor phenotype.

6. Page 11: The last line cut off before reference [26]; Figure 5D. This panel is not mentioned in the results.

Apologies for this oversight, have included Figure 5D (now figure 4D) in the text of the Results section.

Reviewer #2 (Recommendations for authors):General recommendations that span multiple experiments/figures:– Injection sites should be documented for each experiment, using a second virus or marker to document viral spread and actual targeting.

This is an important consideration for all experiments, particularly the behavioral assays. Although we cannot include a second virus/marker retrospectively to our experiments, we have included supplementary figures illustrating all of the brain injection sites that were used in the behavioral experiments. This is included as a figure supplement to figure 7 (now figure 6) (Figure 6 figure supplement 1).

– Explanation is needed as to why all spinal cord injections seem to be at thoracic level 13 (T13), yet L2-L4 may make more sense given the hind limb behavioral phenotypes being assessed.

The spinal cord injections were made below the T13 vertebra, which corresponds to the L3 segment of the spinal cord. In the caudal spinal cord the alignment of vertebrae to spinal cord segments do not correspond exactly, we have added this explanation to the methods section of the manuscript (page 23 line 18).

– What the authors consider nucleus raphe magnus (NRM) should be clarified (e.g. Bregma levels). The presented tissue sections appear fairly caudal and will include multiple subtypes of serotonergic neurons, one of which may innervate the dorsal spinal cord and another that innervates the ventral spinal cord motor centers and likely influences motor gain. The more caudal, the greater enrichment for the latter subtype. It is likely that multiple functionally distinct subtypes of serotonergic neurons are being manipulated in the hM3Dq experiments targeting the presumed NRM. This should be considered, as well as published work in this area should be referenced. It is particularly relevant because if the presumed NRM injections are too caudal, most of the midline serotonergic cells would be those projecting to the ventral (not dorsal) horn and indeed this is what they see in Figure 4.

See above

– References that should be included as relates to medullary serotonergic neuron subtypes, functions, and hodology: Iceman, K.E. et al., 2013 J. Neurophysiol; Richerson, G.B. 2001 Respir. Physiol.; work of Gordon Mitchell; Brust et al., 2014 Cell Reports; Hennessy et al., 2017, J. Neuroscience; Depuy, S.D., et al., 2011 J. Neurosci; work of Peggy Mason.

Thank you for drawing out attention to these studies. The experiments in which midline neurons were labelled/activated will likely also influence the neurons described in these studies. We have expanded our discussion to include medullary serotonergic neurons that regulate respiration and breathing and have included the suggested references (page 19 line 11). We have also included these papers when discussing the results of our electrophysiological experiments (page 20 line 8).

– Suggest greater caution around the use of terms "antinociceptive" and "pronociceptive" (e.g. in the title). Are the behavioral tests performed at pain thresholds? More information for the reader is needed here.

The read-outs of our sensory tests were withdrawal responses which are in general accepted to indicate nociceptive (“pain”) thresholds. Although we cannot, in all cases, be absolutely certain that a withdrawal response indicates a painful sensation, we feel the terms anti- and pro-nociceptive are justified. We would therefore like to maintain the wording in the title.

– More details should be supplied around mouse lines: are the Cre driver lines BAC transgenics, IRES-cre knock-ins, or knock-in/knock-out lines? References are provided, but a direct provision of details would be helpful to the reader.

The TPH2::cre is a BAC transgenic from the Jackson laboratory and the HoxB8-cre is a BAC transgenic mouse line generated in house. We have included a description of this in the methods section to make the generation of these animals clearer (page 23 line 6).